# Speciation of organic aerosols in the Saharan Air Layer and in the free troposphere westerlies

M. Isabel García[1,2], Barend L. van Drooge[3], Sergio Rodríguez[1] and Andrés Alastuey[3]

[1] Izaña Atmospheric Research Centre, AEMET, Joint Research Unit to CSIC "Studies on Atmospheric Pollution", Santa Cruz de Tenerife, 38001, Spain
[2] Department of Chemistry (T.U. Analytical Chemistry), Faculty of Science, University of La Laguna, La Laguna, 38206, Spain
[3] Institute of Environmental Assessment and Water Research, CSIC, Barcelona, 08034, Spain

*Correspondence to*: Sergio Rodríguez (srodriguezg@aemet.es)

**Abstract.** We focused this research on the composition of the organic aerosols transported in the two main airflows of the subtropical North Atlantic free troposphere: (i) the Saharan Air Layer – the warm, dry and dusty airstream that expands from North Africa to the Americas at subtropical and tropical latitudes – and (ii) the westerlies – which flow from North America

over the North Atlantic at mid and subtropical latitudes –. We determined the inorganic compounds (secondary inorganic species and elemental composition), elemental carbon and the organic fraction (bulk organic carbon and organic speciation) present in the aerosol collected at Izaña Observatory, ~2400 m a.s.l.. on Tenerife Island. The concentrations of all inorganic and almost all organic compounds were higher in the Saharan Air Layer than in the westerlies, with bulk organic matter concentrations within the range 0.02–4.0 µg·m$^{-3}$. In the Saharan Air Layer, the total aerosol population was by far dominated

by dust (93% of bulk mass), which was mixed with secondary inorganic pollutants (<5%) and organic matter (~1.5%). The chemical speciation of the organic aerosols (levoglucosan, dicarboxylic acids, saccharides, n-alkanes, hopanes, polycyclic aromatic hydrocarbons and those formed after oxidation of α-pinene and isoprene, determined by gas-chromatography coupled with mass-spectrometry) accounted for 15% of the bulk organic matter (determined by the thermo-optical transmission technique); the most abundant organic compounds were saccharides (associated with surface soils), secondary

organic aerosols linked to oxidation of biogenic isoprene (SOA ISO) and dicarboxylic acids (linked to several primary sources and SOA). When the Saharan Air Layer shifted southward, Izaña was within the westerlies stream and organic matter accounted for ~28% of the bulk mass of aerosols. In the westerlies, the organic aerosol species determined accounted for 64% of the bulk organic matter, with SOA ISO and dicarboxylic acids being the most abundant; the highest concentration of organic matter (3.6 µg·m$^{-3}$) and of some organic species (e.g. levoglucosan and some dicarboxylic acids) were associated

with biomass burning linked to a fire in North America. In the Saharan Air Layer, the correlation found between SOA ISO and nitrate suggests a large-scale impact of enhancement of the formation rate of secondary organic aerosols due to interaction with anthropogenic NO$_x$ emissions.

## 1 Introduction

Atmospheric aerosols, or particulate matter, have an influence on processes affecting climate, on continental and marine ecosystems, and on human health. The magnitude of these effects depends on aerosols composition, which may include secondary inorganic species (e.g. sulphate, nitrate, ammonium and sea salt), mineral dust, elemental carbon and a number of organic species constituting the so-called organic aerosols (OA) (IPCC, 2013). OA accounts for an important fraction of particulate matter, ranging from ~20% (continental mid-latitudes) to ~90% (tropical forested areas) (Kanakidou et al., 2005). As with other aerosol components, OA also contributes to (i) light scattering and absorption (Kirchstetter and Novakov, 2004), (ii) cloud formation providing cloud condensation and ice nuclei (Sun and Ariya, 2005), and (iii) heterogeneous chemical reactions in the atmosphere (Kanakidou et al., 2005).

Principal sources of Primary OA (POA) include vegetation, fossil fuel combustion, biomass burning, biological aerosols and particles from soils. Precursors of Secondary OA (SOA) include natural and anthropogenic sources (Volkamer, et al., 2006; Gouw and Jimenez, 2009); emissions of biogenic volatile organic compounds (VOCs) contribute significantly to the global budget of SOA (Guenther et al., 2012). Some important factors influencing SOA formation are reactive nitrogen species ($NO_x$) (Presto et al., 2005; Ng et al., 2007, 2008), which are further oxidized to the highly reactive nitrate radical ($NO_3$). $NO_3$ interacts with VOCs in gas-phase, likely having an impact on global OA levels as indicated by modelling (Pye et al., 2010) and experimental work (Surratt et al., 2006). In daytime, $NO_x$ can react with organic peroxy radicals ($RO_2$) resulting in peroxy nitrates ($RO_2NO_2$) and alkyl and multifunctional nitrates ($RONO_2$) (O'Brien et al., 1995); the formation of organic nitrates provisionally sequesters $NO_x$, which can suffer long-range transport to more remote environments (Horowitz et al., 1998; Mao et al., 2013). At nighttime, the VOCs-$NO_3$ interaction dominates, with SOA yields greater than those for OH or $O_3$ oxidation (Ng et al., 2017 and references therein). Previous modeling studies carried by Hoyle et al. (2007) suggested that, during twilight conditions, ~ 21% of the global average SOA may be due to oxidation of SOA precursors by $NO_3$, and measurements performed by Brown et al. (2009) found that, during nighttime, 1-17% of SOA was the result of NO3 initiated isoprene oxidation. In remote environments, VOCs enhance condensational growth of new particles, which can enter the free troposphere (FT) by means of elevated mountains (García et al., 2014). These tropospheric aerosols are subject to much greater lifetimes and wind speed than in the planetary boundary layer (BL), favouring long-range atmospheric transport and aerosol impacts (Winker et al., 2013). The aged and processed long-range transported OA is of particular interest, and is spatially representative of the remote background conditions having important implications for global air quality and climate.

The most extended technique used to quantify the amount of bulk organic and elemental carbon in the atmospheric aerosols is the thermo-oxidant combustion and optical detection (Birch and Cary, 1996; Cavalli et al., 2010; Karanasiou et al., 2015). This is a useful method for mass closure, but does not provide information on OA speciation and consequently on OA

sources and properties related to impacts. Alternatively, gas-chromatography coupled with mass-spectrometry analysis of aerosol samples allows the speciation of the organic compounds and the quantification of many of those identified as tracers used to distinguish sources and processes contributing to the budget of OA (Bauer et al. 2008; Mazurek et al. 1989; Claeys et al., 2007; Hallquist et al., 2009; Howsam and Jones 1998; Iinuma et al., 2007; Kawamura and Kaplan, 1987; Medeiros and

5 Simoneit 2007; Narukawa et al., 1999; Rogge et al., 1993; Shauer et al. 2002; Simoneit et al. 1991; Simoneit et al. 2004a; Simoneit, 2002; Szmigielski et al., 2007). A number of studies have focused on OA speciation in urban areas (Alier et al., 2013; Kawamura and Kaplan, 1987; Puxbaum et al., 2007; Schauer et al., 1996; Simoneit et al., 1991; Van Drooge and Grimald, 2015) compared to remote environments. Studies in the free troposphere are less common (Simoneit et al. 2004b; Fu et al., 2008, 2014; Wang et al., 2009; van Drooge et al., 2010; Meng et al., 2014), in spite of the fact that they are of

10 interest due to the long-range transport potential linked to the high wind speeds above the boundary layer.

In this study we focused on the OA transported from the inner Sahara over the North Atlantic in the so-called Saharan Air Layer (SAL); Prospero and Carlson, 1972). In summertime, the continental BL depth grows up to 5 km a.s.l. over the Sahara (Cuesta et al., 2009) and the prevailing easterly winds prompt the export of warm Saharan air to the North Atlantic above the cool NNE trade winds that blow in the marine boundary layer. This results in the development of the SAL – a warm, dry and

15 stable air stream that expands from the North African coast, at altitudes 2 to 5 km a.s.l., to the Americas (Prospero and Carlson, 1972; Tsamalis et al., 2013) –. Because of the high stability associated with the warm air above the cool marine air, the SAL acts as a band conveyor that transports continental Saharan dusty air – originally placed near ground – over the North Atlantic; in addition to dust, other substances such as pollutants, vegetation debris or microorganisms are carried mixed with dust.

OA in the SAL has received little attention, even if its impacts are of interest. Anthropogenic bioaccumulative and toxic organic compounds (including organochlorine and organophosphate pesticides, polycyclic aromatic hydrocarbons, polychlorinated biphenyl) are transported from the Western Sahara to the Caribbean within the SAL (Garrison et al., 2013). Viruses, bacteria, fungi and pollens also travel mixed with dust across the Atlantic (Griffin et al., 2007; Izquierdo et al., 2011). Field measurements in the SAL at Izaña Observatory found dust to be the major ice nuclei at temperatures colder than

25 -30ºC (Boose et al., 2016), whereas the observed ice nuclei at -8ºC points to a role of OA as ice nuclei at warm temperatures (Conen et al., 2015).

In this study we primarily focused on the origin of OA in the SAL. We collected in-situ aerosol samples directly into the high-altitude SAL at Izaña Observatory, located at ~2400 m a.s.l. on Tenerife Island. The profile of the organic species was used for source apportionment of the bulk organic matter. The results were compared with a similar data set obtained during

the same campaign under the westerlies (WES) airflow that regularly brings air from North America across the North Atlantic. The observed differences illustrate the diversity of OA sources over the North Atlantic free troposphere.

## 2 Methodology

### 2.1 Sampling site

Sample collection was performed at the Izaña Global Atmospheric Watch (GAW) Observatory on Tenerife Island (Fig. 1; 16º29'58"W, 28º18'32"N). The site is located on a mountaintop (2373 m a.s.l), surrounded by pine forest (whose limits lie

between ~500–2300 m). The Observatory remains almost permanently above the temperature inversion layer associated to the trade winds, which separates the moist marine BL from the dry FT avoiding vertical mixing before sunrise. Sunlight during daytime activates thermal convection, developing orographic thermal-buoyant upslope winds, that transport species emitted in the BL by biogenic and anthropogenic sources (see details in Rodríguez et al., 2009).

### 2.2 Sampling

Samples were collected within the Izaña Observatory annual aerosols summer campaign in August 2013. Particulate Matter (PM) was collected on pre-heated (at 205ºC) quartz filters (Pall Science 150 mm diameter) on high volume air samplers (Hi-Vol; MCZ) at a flow rate of 30 $m^3 \cdot h^{-1}$. 30 samples of total particulate matter ($PM_T$) were collected daily during night-time (22:00–6:00 GMT; FT) and 12 samples of particulate matter smaller than 2.5 µm ($PM_{2.5}$) on non-consecutive days during daytime (10:00–16:00 GMT; BL). Field blanks were collected weekly and treated like the samples regarding preparation,

transport and storage, as part of the quality assurance / quality control (QA/QC) protocol.

### 2.3 Chemical Analysis

### 2.3.1 Organics

A quarter of the filter sample was used for the organic compounds speciation by gas-chromatography coupled with mass-spectrometry (GC-MS). A detailed description of the analytical method is given elsewhere (Fontal et al. 2015, van Drooge

and Grimalt 2015). Briefly, filters were spiked with deuterated standards of acids, anhydro-saccharides, alkanes, and polycyclic aromatic hydrocarbons (PAHs), and extracted ultrasonically in a mixture of dichloromethane and methanol. Extracts were filtered and concentrated to 0.5 ml. For the analysis of polar compounds, i,e. acids and saccharides, a 25 µL aliquot of the extract was evaporated to dryness, and 25 µL of bis-(trimethylsilyl)-trifluoroacetamide (BSFTA) + trimethylchlorosilane (99:1) (Supelco, Bellefonte, PA, USA) and 10 µl of pyridine (Merck, Darmstadt, Germany) were

added and left overnight to derivatize the polar compounds to their trimethylsilyl esters and ethers for analysis by GC-MS. The remaining extract was used for the analysis of PAHs, n-alkanes and hopanes, and was cleaned up by adsorption column chromatography, packed with 1 g of aluminium oxide (Merck, Germany). The analytes were eluded with 10 ml of hexane: dichloromethane (1:2 (v/v), Merck, Germany), which was collected and concentrated to 1 ml by rotovap and to 25 µl under a gentle nitrogen stream for quantification by GC-MS (Thermo Trace GC Ultra – DSQ II). The MS detector was operated in

full scan (m/z from 50 to 650) and electron impact (70 eV) ionization mode for the polar compounds. The sample extracts for the analysis of non-polar species were performed in selected ion monitoring (SIM) mode for the corresponding ions of the compounds. Organic species were identified by their GC retention time and characteristic ions in the MS (see section S1 of the supplement).

### 2.3.2 OC and EC

Organic and elemental carbon (OC and EC) were analysed by thermal-optical transmittance (TOT, Sunset Laboratory Inc. ™) by using the EUSAAR2 protocol (Cavalli et al., 2010). The method provided four OC fractions (OC1, OC2, OC3 and OC4), the more volatile of which were discarded based on the results of the field blank filters analysis. Organic Matter (OM) was determined using the ratio OM/OC = 1.8 for remote places (Pitchford et al. 2007; Dzepina et al. 2015).

### 2.3.3 Inorganics

The methodology used for the inorganic speciation is described in detail in Rodríguez et al. (2015). Briefly, soluble species were determined by ion chromatography ($SO_4^=$, $NO_3^-$, $Cl^-$) and selective electrode ($NH_4^+$) after water leaching a fraction of filter. Elemental composition was determined by Inductively Coupled Plasma Atomic Emission Spectrometry (ICP-AES, IRIS Advantage TJA Solutions, THERMO™) and Inductively Coupled Plasma Mass Spectrometry (ICP-MS, X Series II, THERMO™) after acid digestion of the sample. Mineral dust was calculated as the sum of $Al_2O_3$ + $SiO_2$ + Fe + $CaCO_3$+ K + Na + Mg + P + Ti + Sr (see details in Rodríguez et al., 2011, 2015) and normalized so Al accounts for 8% of the dust mass (see details in Pérez García-Pando et al., 2016). $SO_4^=$ was split into non-sea salt sulphate (nss-$SO_4^=$) and sea salt sulphate (nss-$SO_4^=$ = $SO_4^=$ - ss-$SO_4^=$) based on the relation between marine Na and $SO_4^=$. Blank field filters were subject to gravimetric and chemical analysis and mean values subtracted to the $PM_x$ samples.

### 2.4 Meteorology

The air mass origin and transport was tracked by means of backward trajectory analysis. Calculations were performed with the HYbrid Single-Particle Lagrangian Integrated Trajectory Model (HYSPLIT, http://ready.arl.noaa.gov/HYSPLIT.php; Draxler and Rolph, 2003; Draxler and Hess, 1998; Stein et al., 2015) developed by the National Oceanic and Atmospheric Administration (NOAA). HYSPLIT was run with the National Centre for Environmental Prediction's (NCEP) Global Data Assimilation System (GDAS, 1 degree) data set. Ten-day back-trajectories arriving at 2400 m a.s.l were computed daily (00 UTC) for August 2013.

### 2.5 Data treatment

In order to observe the similarities and differences among the chemical composition of the samples, the experimental organic compound data were merged for evaluation with Multivariate Curve Resolution Alternating Least Squares (MCR-ALS). The joint dataset was imported into MATLAB 7.4 (The Mathworks, Natick, USA) for subsequent calculations using MATLAB PLS 5.8 Toolbox (Eigenvector Research Inc, Masson WA, USA) (Jaumot et al. 2005). The MCR-ALS method decomposes the data matrix using an Alternating Least Squares algorithm under a set of constraints such as non-negativity, unimodality, closure, trilinearity or selectivity (Tauler et al., 1995a, b). The MCR-ALS method had been applied successfully in a previous study on organic aerosol in urban and rural areas (Alier et al. 2013; Van Drooge and Grimalt 2015). The variance explained by the different components is similar to a PCA, but not orthogonal. Since the natural sources in the environment are rarely orthogonal, the MCR-ALS method provides more realistic descriptions of the components than the orthogonal database decomposition methods. Multi-linear regression tools were applied to quantify the contribution of the identified sources to the total OM.

## 3 Results and Discussion

We collected aerosol samples in four different airflows: 2 FT airflows and other 2 airstreams potentially mixed with BL air. Samples collected at night ($PM_T$) are representative of the two FT airflows that prevail in this region: the WES and the SAL. As already described, the WES flow from North America across the North Atlantic at mid-latitudes, with their southern edge shifting to the subtropics in winter, and flow over Canada (Merry and Moody, 1996) reaching Izaña after circulation around the Azores High in summer (see back-trajectories of the samples collected from 26 Aug to 30 Aug – with "ddmmm" referring to ending sampling day – in Fig.1A). The SAL expands from North Africa to the Americas at subtropical latitudes in summertime (see back-trajectories associated during the study period in Fig. 1B), the season in which the Izaña Observatory is mostly within this dusty airstream and the presence of the WES is associated with southern shifts of the SAL.

Samples collected during daylight ($PM_{2.5}$) are representative of the FT potentially mixed with BL air, more specifically the BL-SAL mixing and BL-WES mixing. The presence of BL air is associated with the development of buoyant upslope winds caused by the warming of the terrain, which typically results in increases of primary gaseous pollutants and new particle formation at Izaña (García et al, 2014; Rodríguez et al., 2009).

Thus, in this study we differentiate between 4 scenarios: $PM_T$ (FT, night-time) within (i) the SAL (FT-SAL) and (ii) the WES (FT-WES), and $PM_{2.5}$ (BL, daytime) within (iii) the SAL (BL-SAL) and (iv) the WES (BL-WES).

### 3.1 Major Components

Table 1 shows the composition of the aerosol samples collected at Izaña. Concentrations of aerosol major components are the same as those found in previous studies (Maring et al., 2000; Rodríguez et al., 2011). All species present much higher

concentrations within the SAL than within the WES. Some species show slightly higher concentrations during the day linked to the upslope winds and boundary layer air.

## 3.2 Organic molecular tracers

In the next sections, organic speciation results are described. Table 2 shows the average concentrations of the 40 organic
compounds analysed in this study under the different scenarios (SAL and WES) for $PM_T$ and $PM_{2.5}$. In order to improve insight into the origin and sources of some FT organic aerosols, correlations among the organic groups and the major species are evaluated by means of the Pearson correlation coefficient (r) (Table 3); significance levels are determined according to the p-value (p). This coefficient is applied for all correlation throughout the manuscript.

### 3.2.1 Levoglucosan

Levoglucosan (1,6-anhydro-β-D-glucopyranose) is emitted during biomass burning as a consequence of the thermal alteration of cellulose and hemi-cellulose present in vegetation (Simoneit, 2002). It is considered a particle-phase marker for the identification of wood combustion due to its source specific emission, but its atmospheric stability is still a matter of discussion. Experiments carried out by Hennigan et al. (2010) and Hoffmann et al. (2010) showed that levoglucosan reacts with gas-phase hydroxyl radicals (OH), especially under high relative humidity conditions. However, studies performed by
Fraser and Lakshmanan (2000) demonstrated no degradation of levoglucosan under acidic conditions over a period of 10 days.

Levoglucosan daily values measured at Izaña within the FT and the BL were < 1.5 ng·m$^{-3}$ for all samples, except on 28 Aug when ~9 ng·m$^{-3}$ was measured. Hysplit back-trajectories and the Atmospheric Infrared Sounder (AIRS) satellite images (NASA) indicate the North American origin of the air masses, where several wildfires – such as the Rim Fire – were
affecting western USA 10 days before the air mass started moving towards Izaña (detailed information not provided for sake of brevity). We detected a long-range transport biomass burning plume within the FT from the fires originated in North America. Other studies performed at Pico Mountain Observatory (Azores, 2225 m a.s.l.) have also detected the impact of other biomass burning plumes by means of levoglucosan detection (Dzepina et al, 2015). These results lend support to the atmospheric stability of levoglucosan, under the specific atmospheric conditions of this long-range transport. Due to the
particular composition of this biomass burning event (BBE), we will discuss this sample separately (Table 2) and the sample will not be included when describing the general composition of the samples collected at night under westerlies conditions (FT-WES; Table 2). Levoglucosan concentration, measured at Izaña during the BBE (9.3 ng·m$^{-3}$), is similar to the average levels detected in the marine BL over the Azores in the North Atlantic (5.2 ng·m$^{-3}$) or at a free tropospheric site in the European continent (7.8 ng·m$^{-3}$), but much lower than those found at sites under the influence of local BB or continental
sites in winter (653–1290 ng·m$^{-3}$) (Puxbaum et al. 2007).

### 3.2.2 Dicarboxylic acids

Dicarboxylic acids can be emitted in small quantities from several natural and anthropogenic primary sources such as vegetation, meat cooking and motor exhaust emissions (Kawamura and Kaplan, 1987, Narukawa et al., 1999), although atmospheric photochemical transformation of volatile and semi-volatile organic compounds is considered to be an important source for the presence of these aged compounds in the atmosphere (Alier et al. 2013, Jang and McDow, 1997; Kleindienst et al. 2012; Paulot et al. 2011). This oxidative degradation of VOCs by tropospheric oxidants may be responsible for the similar mean ∑dicarboxylic acid concentrations within the FT (SAL: 14.4 ng·m$^{-3}$; WES: 16.7 ng·m$^{-3}$) and the BL (SAL: 11.1 ng·m$^{-3}$; WES: 12.9 ng·m$^{-3}$) at Izaña.

Succinic (suc) and phthalic (pth) acids were the most abundant dicarboxylic acids (Table 2) with FT and BL average values (suc: 6.5–3.7 and pth: 3.2–2.8 ng·m$^{-3}$ for the FT-BL) much lower than those found for PM$_{2.5}$ in the FT Mount Tai (suc: 30 ng·m$^{-3}$ ; Wang et al., 2009), similar to those observed in PM$_T$ in the Himalayas (4276 m a.s.l.) (suc: 13.7 and pth: 9.5 ng·m$^{-3}$; Cong et al., 2015), but higher than those detected for PM$_T$ in the North Pacific remote marine (suc: 2.8 and pth: 0.66 ng·m$^{-3}$; Kawamura and Sakaguchi., 1999). Malic acid within the BL (the third most abundant poly-acid at Izaña; Table 2) might be photochemical in origin via OH oxidation of the surrounding biogenic compounds transported by the daytime upslope winds. The Izaña Observatory is surrounded downhill by a forest ring – an important source of biogenic volatile organic compounds (BVOCs) – which contributes significantly to the concentrations measured at Izaña. Oxidation of these biogenic precursors may also provide important amounts of C$_7$–C$_9$ dicarboxylic acids.

High concentrations of dicarboxylic acids have been reported in plumes from BB (Narukawa et al., 1999; Gao et al., 2003), which is in line with the observed values for the long-range transport BBE (82 ng·m$^{-3}$). Concentrations of succinic, glutaric and malic acids were high (~33 ng·m$^{-3}$, ~7 ng·m$^{-3}$ and ~32 ng·m$^{-3}$, respectively; Table 2), compared to the rest of the period, most likely as a consequence of the lofted concentration emitted in the open fire and long-range transport photochemical aging processes.

### 3.2.3 Saccharides

Primary saccharides and polyols are tracer compounds of surface soils (Medeiros and Simoneit 2007; Simoneit et al. 2004a), related to plant tissue and microorganisms. Glucose (α, β), fructose and sucrose are important constituents of OM in soils (Simoneit et al. 2004a), whereas mannitol is related to airborne fungal spores (Bauer et al. 2008). They are completely water-soluble, contributing to water soluble organic carbon (WSOC) in aerosols (Simoneit et al. 2004a). Wind erosion and up-lifted soil dust emit these compounds to the atmosphere (Simoneit et al. 2004a).

The average concentration of the saccharides exhibits a marked difference within the FT-PM$_T$ (23.5 ng·m$^{-3}$) and the BL-PM$_{2.5}$ (3.5 ng·m$^{-3}$). Previous studies have shown that some organic compounds are strongly particle-size dependent (Mochida

et al., 2003; Van Drooge and Grimalt, 2015), with special emphasis on sugars and sugar alcohols, which are present mostly in very large particles (Graham et al., 2003; Yttri et al., 2007). This size-segregation is clearly seen for the $PM_{2.5}$ samples collected within the BL, that cuts off an important fraction of the coarse organic soil dust aerosol, showing similar concentrations under SAL and WES influence (BL-SAL = 3.3 ng·m$^{-3}$; BL-WES = 4.0 ng·m$^{-3}$). A different scenario takes place with the $PM_T$ samples, for which concentrations rise one order of magnitude from SAL influence to clean conditions (FT-SAL = 27 ng·m$^{-3}$; FT-WES = 2.5 ng·m$^{-3}$) linked to Saharan dust contribution. The average saccharide levels in the FT-SAL are higher than those observed in a natural forest area in tropical India (12.78 ng·m$^{-3}$; Fu et al., 2010) and a rural background in Norway (10.4 ng·m$^{-3}$; Yttri et al., 2007), but very similar to the average concentrations measured in the FT over central China (28.1 ng·m$^{-3}$; Fu et al., 2014).

Glucose (α+β) was the predominant saccharide within the SAL, with a mean FT concentration of ~10 ng·m$^{-3}$ (Table 2); both isomers showed a statistically significant correlation (r = 0.99, p<0.01) consistent with their relation in the soil (Simoneit et al. 2004a). Under the WES airflows, glucose (α+β), sucrose and mannitol were slightly higher during the day (BL; Table 2), suggesting that there might be some soil contribution of transported terrestrial OM by land breeze. This load is more evident in the sucrose (FT-WES = 0.3 ng·m$^{-3}$; BL-WES = 1.3 ng·m$^{-3}$; Table 2), which is a predominant sugar in the phloem of plants playing a key role in developing flowers (Bieleski et al., 1995) and has been suggested as a tracer for airborne pollen grains (Fu et al., 2012).

### 3.2.4 n-Alkanes

n-Alkanes, or aliphatic hydrocarbons, are a result of biogenic and anthropogenic emissions such as plant waxes and fossil fuel combustion products (Mazurek et al. 1989; Simoneit et al. 1991; Shauer et al. 2002). In the present study n-alkanes from $nC_{24}$ to $nC_{34}$ were quantified, with total n-alkane mean concentrations (~8 ng·m$^{-3}$ in the FT and in the BL) much lower than those measured in the tropical Indian summer (126 ng·m$^{-3}$; Fu et al., 2010), but similar to those found in rural Spain during the warm period (12 ng·m$^{-3}$; Van Drooge and Grimalt, 2015).

Information about the possible source may be provided by the carbon number maximum ($C_{max}$). In general, $nC_{27}$, $nC_{29}$ and $nC_{31}$ are related to waxes from terrestrial higher plants, whereas low-molecular-weight alkanes ($C_{22}$–$C_{25}$) are more associated with combustion sources (Mazurek et al. 1989). At Izaña, the most abundant n-alkanes within the FT-SAL were $nC_{27}$, $nC_{29}$ and $nC_{31}$ (~1.0 ng·m$^{-3}$, ~1.4 ng·m$^3$ and ~1.8 ng·m$^{-3}$ respectively; Table 2) reflecting a vegetative source as previously described for Saharan dust samples measured in the North Atlantic (Simoneit et al., 1977), whereas the BL $nC_{24}$–$nC_{25}$ presented higher concentrations (Table 2) linked to anthropogenic emissions carried by the upslope winds. Another indicator that can be used to show the source type is the carbon preference index (CPI = ∑odd n-alkanes / ∑even n-alkanes) with CPI > 1 related to biogenic origin and CPI ≈1 to combustion processes (Mazurek et al. 1989; Simoneit, 2002). In this study, CPI values ranged from 0.9 to 6.3 with average values for the FT-SAL (~2) higher than those for the BL (~1.7) and the FT-WES

(~1.2), reflecting the greater influence of vegetation within the FT-SAL and the predominance of combustion contribution within the BL and FT-WES samples. Although the vegetative source dominates in the FT, there is a statistically significant correlation between n-alkanes and $NO_3^-$ (r = 0.8, p<0.01; Table 3), mostly due to its anthropogenic fraction ($C_{24}$–$C_{25}$).

### 3.2.5 Hopanes

Hopanes (17α(H),21β(H)-29-norhopane and 17α(H),21β(H)-hopane) are linked to mineral oil and related to unburned lubricating residues from primary vehicle emissions (Rogge et al., 1993; Schauer et al., 1996, 2002). ∑Hopanes mean concentrations were 0.13 and 0.08 ng·m$^{-3}$ within the FT and BL respectively, values much higher than the $7 \cdot 10^{-4}$ ng·m$^{-3}$ measured by von Schneidemesser et al. (2009) in remote Greenland (3200 m a.s.l.) where anthropogenic emissions in the surrounding region are minimal. Under the WES airflows, hopanes concentrations were slightly higher during the day,

suggesting an influence of pollution transported within the BL, related to motorized vehicle emissions. Quantified hopane and norhopane showed a statistically significant correlation (r = 0.97, p<0.01), implying the same emission sources.

A statistically significant correlation is observed in the FT between hopanes and $NO_3^-$ (r = 0.8, p<0.01; Table 3) suggesting that the origin of most $NO_3^-$ in the FT lies in on-road vehicle emissions rather than industry. Anthropogenic sources of $NO_x$ (the major $NO_3^-$ precursor) include fossil fuel emitted from agriculture, power plants, industry and transport. The latter

accounts for almost 50% of nitrogen oxides emissions ([http://www. eea.europa.eu/data-and-maps/indicators/eea-32-nitrogen-oxides-nox-emissions-1/assessment.2010-08-19. 0140149032-3](http://www.eea.europa.eu/data-and-maps/indicators/eea-32-nitrogen-oxides-nox-emissions-1/assessment.2010-08-19.0140149032-3)), with on-road transport in 2010 being the highest (25.2 Tg·year$^{-1}$) compared to non-road (10.1 Tg·year$^{-1}$), shipping (16.2 Tg·year$^{-1}$), aviation (3.0 Tg·year$^{-1}$) or rail (1.6 Tg·year$^{-1}$) (Yan et al., 2014).

### 3.2.6 Polycyclic aromatic hydrocarbons

Polycyclic aromatic hydrocarbons (PAH) are organic pollutants generated during incomplete combustion of organic natural material (e.g. forest fires, volcanic activity) and anthropogenic (e.g. fossil fuel combustion, coke production) sources (Howsam and Jones 1998; Iinuma et al., 2007; Rogge et al., 1993; Schauer et al., 1996, 2002). PAHs are composed of two or more fused aromatic rings and some of them have carcinogenicity, genotoxicity, and are potentially endocrine disruptive, affecting human health. At Izaña mean values of the total PAHs exhibited higher values in the BL (24 pg·m$^{-3}$) than in the FT

(16.3 pg·m$^{-3}$), reflecting the contribution of the upslope winds as described for other organic compounds.

Similar PAHs concentrations were previously found by van Drooge et al. (2010) who measured an average PAH concentration of 33.1 pg·m$^{-3}$ at Izaña. In general, all individual PAHs decreased in concentration with the exception of benz(a)anthracene which increased by a factor of 1.5 and 1.35 with respect to the mean concentrations of the FT and BL correspondingly. Much higher concentrations have been reported in other remote FT locations such as Mt. Tai (1534 m

a.s.l.) where ~9 ng·m$^{-3}$ were measured (Fu et al., 2008). In the FT there is a statistically significant correlation between PAHs and EC (r = 0.7, p<0.01; Table 3), which points to the incomplete combustion of fossil fuels (Frenklach., 2000).

During the detected North America wildfire event (28 Aug), PAH concentration was 9.4 pg·m$^{-3}$, which is much lower than levels measured in Thailand for PM$_T$ samples during BBEs in the dry season (1150 to 4140 pg·m$^{-3}$; Chuesaard et al., 2014).
The concentrations of PAHs measured in the sample corresponding to the fire event were no higher than those observed in the other samples, which may be due to photochemical transformations of PAH in the atmosphere during long-range transport.

### 3.2.7 Tracers of α-pinene oxidation (SOA PIN)

Vegetation emits large quantities of biogenic volatile organic compounds (BVOCs) into the atmosphere compared to anthropogenic VOCs (Guenther et al., 2006, 2012; Lamarque et al., 2010), particularly monoterpenes and isoprene. The most abundant volatile monoterpene, emitted mainly by coniferous trees (i.e. *Pinus canariensis*) is α-pinene (Andreani-Aksoyoglu and Keller, 1995; Rinne et al., 2009; Smolander et al., 2014) and the tracers related to its photochemical oxidation (SOA PIN) are *cis*-pinonic acid, 3-hydroxyglutaric acid (3-HGA) and 3-methyl-1,2,3-butanetricarboxylic acid (MBTCA) (Claeys et al., 2007; Szmigielski et al., 2007).

SOA PIN organic tracers were not detected in all samples, with values in the FT influenced by a few extreme points that increased their average concentration. SOA PIN exhibited the lowest concentration in the FT-WES (1.2 ng·m$^{-3}$) with a predominance of *cis*-pinonic acid (Table 2). Aircraft measurements in the FT over central China (Fu et al., 2014) recorded higher concentrations of 3-HGA (8.5 ng·m$^{-3}$) and MBTCA (1.9 ng·m$^{-3}$) than those measured in the present study (Table 2). Further generation oxidation products (3-HGA and MBTCA) were higher in the BL (0.51 and 0.24 ng·m$^{-3}$ correspondingly; Table 2), with a statistically significant correlation (r = 0.90, p<0.01) pointing to a same precursor. Monoterpenes react relatively rapidly, with atmospheric lifetimes ranging from minutes to hours (Saxton et al., 2007), resulting in α-pinene emitted in the forest ring that reacts along its upward transport to the Observatory. Daytime emissions of gaseous α-pinene at Izaña were measured by Fisher at al. (1998) with concentration in the range of 0.011–0.102 ppbv (mean: 0.028 ppbv), supporting evidence of its origin being close to the Observatory during the day.

### 3.2.8 Tracers of isoprene oxidation (SOA ISO)

It is estimated that about a half of the total global BVOCs emission is due to isoprene (535 Tg·y$^{-1}$; Guenther et al., 2012), making it the largest BVOC emitted from land vegetation (Guenther et al., 2006). Isoprene emission is limited to a number of species in the plant kingdom, contrary to many other BVOCs that are emitted from most plants (Guenther et al., 2012). Secondary products of isoprene oxidation (SOA ISO) evaluated in the present study are 2-methylglyceric acid (2-MGA), 2-methylthreitol (2-MT1) and 2-methylerythritol (2-MT2) (Claeys et al., 2004; Hallquist et al., 2009).

Analogous FT concentrations of 2-methylthreitol and 2-methylerythritol were measured in the present study under the SAL (7.3 and 16.8 ng·m$^{-3}$; Table 2) and over the central China FT (8 and 17 ng·m$^{-3}$; Fu et al., 2014), one of the most important source regions of isoprene emission in the world during summertime (Guenther et al., 1995). Similar SOA ISO concentrations were found in the BL within the SAL and the WES (~17 and ~16 ng·m$^{-3}$ respectively) revealing the emission

and subsequent ascending transport of biogenic or anthropogenic compounds, as found in previous studies performed at Izaña, which observed emissions of isoprene during daytime associated with anthropogenic compounds (Salisbury et al., 2006). A statistically significant correlation among 2-MT1 and 2-MT2 was found for individual values (r = 0.90, p<0.01) as previously observed in other studies (Ion et al., 2005; El Haddad et al., 2011), but with a mass concentration ratio 2-MT1 vs. 2-MT2 (slope from linear regression=2.3) slightly lower than that found by El Haddad et al. (2011). This statistically

significant correlation between the two diastereoisomers would seem to indicate they formed through the same photo-oxidation process.

The highest concentration of SOA ISO was measured under the FT-SAL (~28 ng·m$^{-3}$), associated with Saharan dust, as was observed for SOA PIN (~33 ng·m$^{-3}$). However, global estimations of isoprene and α-pinene emissions and sources show they are diverse and not equally distributed around the globe (Luo et al., 2010; Guenther et al., 2012; Sindelarova et al., 2014).

The correlation between total concentration of SOA ISO and total concentration of SOA PIN (Fig. 2) exhibits two distinct trends in the FT that might be associated with different global sources of the precursor volatile compounds, although the trajectories of the sampled air mass do not clearly distinguish between different origins. Some species with high isoprene emission potential have been identified in central and western Africa, but quantification of isoprene emissions are largely unverified for West Africa (Saxton et al., 2007). Several evaluations of isoprene and α-pinene global emissions (Luo et al.,

2010; Guenther et al., 2012; Sindelarova et al., 2014) confine the North Africa sources to a small belt over the northern part of Morocco, Algeria and Tunisia, whereas Europe is a potential source. Some episodes, for which SOA PIN and SOA ISO were measured, do not have a trajectory over this African belt – based on the Hysplit model – , suggesting that air masses from Europe can also incorporate gaseous precursors and oxidized species previous to their passing over Africa and Izaña in the Atlantic Ocean.

Rodríguez et al. (2011) previously reported high concentrations of SO$_4^=$, NO$_3^-$ and NH$_4^+$ for air masses arriving at Izaña from the Atlantic coast of Morocco, Eastern Algeria, Northern Algeria and Tunisia. Industrial states with sources of gaseous precursors (SO$_x$, NO$_x$ and NH$_3$) of these aerosol compounds were identified. Although African emissions seem to be responsible for the pollutants reaching the North Atlantic FT, Europe may also contribute with amounts of pollutants that should not be neglected as evidenced by the concentration of SOA PIN and SOA ISO arriving at Izaña; this is supported by a

previous study (Garrett et al., 2003) in which it was suggested that carbonaceous, sulphate, and nitrate particles – in aerosol plumes transported from North Africa over the North Atlantic Ocean within the FT – were anthropogenic pollution from Europe. These species may play a key role in secondary organic aerosol formation, as some studies point to the influence of

anthropogenic emission on secondary organic aerosol formation (El Haddad et al., 2011; Hoyle et al., 2011). SOA ISO seems to depend heavily on the conditions (aerosol acidity, $NO_x$ concentrations and pre-existing aerosol) used to oxidize isoprene (Surratt et al., 2006, 2007; Marais et al., 2016). $NO_x$ concentration determines the pathway (low-$NO_x$ and high-$NO_x$) followed by the isoprene oxidation, leading to different secondary organic species (Paulot et al., 2009a, b); the low-$NO_x$ pathway is ~ 5 times more efficient than the high-$NO_x$ pathway (Marais et al., 2016). Experiments carried out by Kroll et al. (2006) evidence how isoprene SOA yield varies depending on $NO_x$ concentration, increasing from no injected $NO_x$, to a plateau between 100 and 300 ppb $NO_x$, and decreasing at higher $NO_x$ concentrations.

We found the relation among SOA ISO and $NO_3^-$ within the FT-SAL (Fig.3) presents three tendencies which might be associated with the ratio isoprene:$NO_x$ in the source. The different correlations are supported by the fact that the SOA ISO markers (2-MTs and 2-MGA) do not exhibit the same temporal trend (r = 0.4, p<0.05 within the FT-SAL) (FT-SAL 2-MTs us 2-MGA-$r^2$: 0.1), which has been suggested to be linked to the $NO_x$ concentration influence on these SOA ISO markers formation pathways (El Haddad et al., 2011). The high-$NO_x$ pathway leads to the reaction of isoprene peroxy radicals ($iRO_2$) with NO resulting in carbonyl and hydroxynitrate production (Surratt et al., 2006), whereas the low-$NO_x$ pathway leads to the reaction of $iRO_2$ with hydroperoxy radicals ($HO_2$) resulting in hydroxyhydroperoxide (iROOH), and carbonyl production to a lesser extent (Carlton et al., 2009; Paulot et al., 2009). This has implications for the abundance of the secondary organic markers from isoprene photo-oxidation (2-MT and 2-MGA): high-$NO_x$ pathway results in the major product 2-MGA and low-$NO_x$ pathway in major products 2-MTs (El Haddad., 2011).

Statistically significant correlations in the FT between biogenic secondary organic compounds and $NO_3^-$ (r SOA PIN–$NO_3^-$ = 0.6, p<0.01; Table 3) and nss-$SO_4^=$ (r SOA ISO–nss-$SO_4^=$ = 0.6, p<0.01; Table 3), point to its formation from the oxidation of their gaseous precursors $NO_x$ and $SO_2$ respectively. Dust transformation in the FT is also evidenced by its statistically significant correlation with SOA PIN (r = 0.7, p<0.01; Table 3) in addition to both saccharides (r = 0.6, p<0.01) and hopanes (r = 0.9, p<0.01) showing that natural and anthropogenic substances might be mixed after aging processes.

### 3.2.9 Fraction determined of OM

Bulk organic carbon (OC) determined for every single day (thermo-optical transmittance method) at Izaña during this study was within the range 0.01–2.20 µg·m$^{-3}$, which is in line with that found in other FT studies (e.g. 1.4 µg·m$^{-3}$ at Qomolangma, Mt. Everest, 4276 m a.s.l. by Cong et al., 2015 and 4 µg·m$^{-3}$ at NW Pacific, 2–6.5 km column by Heald et al., 2005). FT-$PM_T$ OC under SAL conditions (0.77 µg·m$^{-3}$) was higher than under WES (0.52 µg·m$^{-3}$, including the BBE) events. Figure 4 shows the mass closure (sum of the organic species determined by speciation) of bulk organic matter OM (determined as OC·1.8); this mass closure accounts for 2-100 % of the OM for every single day, depending on the sample and the airflows.

Concentrations of OM were much higher in the FT-SAL (1.39 µg·m$^{-3}$) than under FT-WES (0.04 µg·m$^{-3}$ without the BBE; 0.94 µg·m$^{-3}$ including the BBE) conditions. The selected tracers (levoglucosan, SOA ISO, SOA PIN, n-Alkanes, saccharides, dicarboxylic acids, hopanes and PAHs) represent on average:

(i)  15% of the OM (Fig. 4), under FT-SAL conditions (when mean OM was 1.39 µg·m$^{-3}$). This fraction determined is mainly composed of SOA ISO (30%), saccharides (27%) and dicarboxylic acids (18%).

(ii)  84% of the OM (Fig. 4), in the FT-WES airflows without the BBE (when mean OM was 0.04 µg·m$^{-3}$). This fraction determined is comprised of dicarboxylic acids (44%; mainly succinic and phthalic, indicating aged aerosols after the long-range atmospheric transport), and SOA ISO (34%) with a minor presence of saccharides (8%). Biogenic SOA represent an important fraction of the OM at Izaña (~40%), as seen in other remote high altitude regions (Xu et al. 2015).

(iii)  3% of the OM (Fig. 4) in the FT-WES during the BBE (when mean OM was 3.64 µg·m$^{-3}$). The fraction determined of OM for 28 Aug (3%) contained 68% of dicarboxylic acids (mostly succinic and malic acids) and 8% of the BB tracer levoglucosan. The OM profile of this sample has the highest contribution of aged SOA (di-acids), formed during the long-range atmospheric transport, and the lowest contributions of SOA PIN (4%) and SOA ISO (12%), which may indicate the further oxidation of these products under BBE air masses conditions.

(iv)  64% of the OM in the FT-WES including the BBE (when mean OM was 0.94 µg·m$^{-3}$). This fraction determined is comprised mainly of dicarboxylic acids (50%), and SOA ISO (28%).

Differences in the fraction determined of the samples collected under the FT-SAL and FT-WES influence – as observed in Fig. 4 – is a result of the method limitation, as the analysis of the samples by means of gas chromatography–mass spectrometry (GC-MS) covers a very small fraction (often < 5 %; Alier et al., 2013) of the organic matter. The OM composition determined within the BL (PM$_{2.5}$; BL-FT: 9% of the OM and BL-WES: 77% of the OM; Fig. 4) remains almost the same under both airflows, but with higher concentrations of SOA ISO and n-alkanes under the SAL. However, due to the PM$_{2.5}$ cut-off of the collected particles, the influence of the dust-associated organic compound is likely not well represented as these products are situated in the coarse fraction of the PM.

**3.3 Sources of OM**

We used receptor modelling for apportionment of OM between the OA sources traced by the species included in the speciation performed in this study. This analysis is complementary to the mass closure performed above (Fig.4). The data matrix was decomposed in two factors: loadings (i.e. the relative amount of the chemical compounds in the source) and scores (i.e. the relative contribution of the potential sources to the organic aerosol) (Tauler et al., 2009). The loading factors obtained in the MCR-ALS were used to identify OA sources (Fig.5A1–5C1), whereas the score factors were used as independent variables in the Multi-linear Regression Analysis (MLRA) to apportion the fraction determined of OM between

the identified sources. Three components (sources) were identified (Fig. 5), which accounted for 81% of the total variance. MCR-ALS method was also applied only to $PM_T$ samples to verify the influence of $PM_{2.5}$ in the final results; no significant differences were observed as shown in Fig. S1 of the supplement.

### 3.3.1 Biomass Burning (BB)

The major component (accounting for 63% of the total variance) is associated with levoglucosan, dicarboxylic acids, phthalic acid, SOA PIN, $C_{24}$–$C_{28}$ alkanes and hopanes and PAHs to a lesser extent (Fig. 5A). This factor represents biomass burning aerosols (BB). The peak event in the score factor observed on 28 Aug (Fig. 5A2) is associated with the episode of levoglucosan linked to the long-range transport of BB from North America. SOA PIN indicates photochemical oxidation of biogenic volatile organic compounds during the wild fire. The presence of short-chain dicarboxylic acids, along with large

amounts of malic acid, suggests the effective oxidation of organic species to shorter di-acid chains during long-range atmospheric transport. PAH contribution in this component is low, despite potential emissions of PAHs during biomass burning. The low PAH contributions may be the result of photochemical degradation during long-range transport, which on their term could be related to the presence of higher contributions of phthalic acid in this component. BB in the FT is significantly correlated to the OM (r-FT = 0.40, $p < 0.05$; Table S1) and EC (r-FT = 0.65, $p < 0.01$; Table S1) concentrations.

### 3.3.2 Combustion POA

The second component (accounting for 21% of the total variance) is associated with long-chain dicarboxylic acids, SOA ISO, $C_{24}$ – $C_{29}$ n-alkanes and PAHs (Fig. 5B). This factor is related to the primary organic aerosols linked to combustion sources. This is the component that best represents the variability of $PM_{2.5}$ samples collected during daylight, when the BL may reach Izaña under the slope wind regime. High loadings of suberic ($C_8$) and azelaic acids ($C_9$) indicate the presence of

oxidized compounds in the early stage of photochemical transformation processes, such as the ozonolysis of oleic acid (Moise and Rudich, 2002). Organic species supporting the anthropogenic contribution are lower-molecular-weight n-alkanes ($C_{24}$ – $C_{25}$) and PAHs (from incomplete combustion processes). FT aerosol is also described by this component with the exception of low-molecular-weight alkanes ($C_{24}$–$C_{25}$), which is the main feature of the BL samples. This component is representative of the measured EC for all samples, and representative for the BL, as shown by its statistically significant

correlation (r-All = 0.36 and r-BL = 0.71, $p < 0.05$; Table S1).

### 3.3.3 Organic dust

The third component, accounting for 16% of the total variance, is comprised of short-chain dicarboxylic acids, SOA ISO, saccharides, $C_{26}$–$C_{34}$ alkanes, hopanes, and PAHs to a lesser extent (Fig. 5C). This component, identified as organic dust, is associated with soil re-suspension as evidenced by the saccharides and mannitol high loadings. A major presence of the soil

OM related compounds, those related to fungi and terrestrial higher plants ($C_{27}$, $C_{29}$ and $C_{31}$), suggests fresh and primary OA.

Notwithstanding, glutaric, adipic and pimelic acids indicate oxidation products, suggesting the aging of the samples. As a consequence of this aging, natural and anthropogenic markers are mixed in this component. The biogenic influence is indicated by the presence of soil related markers and oxidation products from isoprene (2MGA, 2MT-1 and 2MT-2), whereas the anthropogenic influence is well defined by the presence of hopanes (primary vehicle emissions) and high molecular weight PAH (products of incomplete combustion). The scores of this component display the highest statistically significant correlation with dust (r-All = 0.84, $p < 0.01$; Table S1) and OM (r-All = 0.64, $p < 0.01$; Table S1) concentrations for all samples. Although this component is not relevant for the BL-PM$_{2.5}$ samples − because part of the compounds are present in the larger particle size fractions − the correlation with dust (r-BL = 0.73, $p < 0.01$; Table S1) and OM (r-BL = 0.75, $p < 0.01$; Table S1) are statistically significant due to the mixing of dust with the anthropogenic compounds.

### 3.3.4 Source apportionment of OM in the SAL and the westerlies

The source apportionment of OM was performed by the multi-linear regression technique described above. The difference between the bulk OM (determined by thermo-optical method) and the sum of the organic species (determined with speciation) was labelled as undetermined fraction. Figure 6 shows the time series of the daily contribution of each source to the OM determined and Fig. 7 shows the average source contribution to total OM in the aerosol samples collected in the different airstreams. The statistically significant correlation between the sum of the three components scores and the OM within the FT (OM r-FT = 0.63, $p < 0.05$; Table S1) indicates that the identified sources might describe, not only the fraction determined of the OM but also the total OM. This significant correlation is not seen for the BL (OM r-BL = 0.33, $p < 0.05$; Table S1), where there could be additional sources.

In the FT-SAL airflow, most OM was undetermined (~85, Fig.7). The three identified sources, i.e. organic dust, combustion POA and biomass burning, accounts for 8%, 6% and 1% of the bulk OM, respectively (62%, 34% and 4% of the OM determined, respectively). The presence of biogenic SOA products mixed with combustion POA was also found in previous studies which suggested that biogenic SOA formation may be more efficient in polluted atmospheres (Gouw and Jiménez., 2009 and references therein).

In the FT-WES, the undetermined fraction accounts for ~36% of the OM (Fig.7). The contribution of the three identified sources, i.e. organic dust, combustion POA and biomass burning, is 22%, 19% and 23% of the bulk OM, respectively (28%, 23% and 49% of the OM determined, respectively). Yttri et al. (2007) proposed biomass burning as a source of saccharides in the OA and Fraser and Lakshmanan (2000) found that some saccharides resist degradation in the atmosphere over a period of 10 days, being able to be transported over long distances; this may be the source of the organic fraction of dust we detected in the FT-WES.

For the BL samples, the dust related component is not well represented (Fig. 6), because the coarse fraction was not sampled here. On the other hand, combustion POA explains 6 and 41 % (Table S2) of the bulk OM for the SAL and WES, respectively. Background regional fires also affect the BL as described by the BB component, which represent 2 and 36 % (Table S2) of the bulk OM for the SAL and WES, respectively.

**4 Conclusions**

The present study focuses on the organic aerosol composition within the two main airflows of the subtropical North Atlantic free troposphere: (i) the Saharan Air Layer – the warm, dry and dusty airstream that expands from North Africa to the Americas at subtropical and tropical latitudes – and (ii) the westerlies – which flows from North America through the North Atlantic at mid and subtropical latitudes –. Atmospheric particulate matter was analysed on secondary inorganic species,

elemental composition, elemental and organic carbon and 40 organic tracer species (levoglucosan, dicarboxylic acids, saccharides, n-alkanes, hopanes, polycyclic aromatic hydrocarbons and those formed after oxidation of α-pinene and isoprene) in order to distinguish possible sources for the organic aerosol. The organic particulate aerosol speciation and its subsequent source apportionment was performed for 42 filter samples collected in summer at the Izaña Observatory (~2400 m a.s.l.) on Tenerife, Spain.

The levels of all inorganic and almost all organic tracers were generally higher under the Saharan Air Layer influence in comparison to the pristine conditions of the Westerlies and the differences in the composition of the organic matter determined under these two air masses were substantial.

In the Saharan Air Layer, the aerosol composition was dominated by dust (93%), secondary inorganic pollutants (<5%) and organic matter (~1.5%). The organic compounds (determined by gas-chromatography coupled with mass-spectrometry)

accounted for 15% of the bulk organic matter and were related to soils (saccharides), biogenic secondary organic aerosols linked to isoprene oxidation (SOA ISO) and natural and anthropogenic primary sources such as vegetation and motor exhaust emissions (dicarboxylic acids).

In the Westerlies, organic matter represented a higher fraction of the total aerosol bulk (~28%) and the organic compounds determined accounted for 64% of the organic matter with dicarboxylic acids and SOA ISO being the most abundant. In this

airstream, a long-range atmospheric transport of a biomass burning plume from North America was detected (with organic matter representing 53% of the total aerosol bulk), supporting the atmospheric stability of levoglucosan over transport and time under certain conditions.

Three sources of organic aerosol, which contribute to the organic matter composition in this part of the North Atlantic, could be resolved in multivariate analysis: one biomass burning-related, one primary combustion-related and one organic dust-

30 related. In the Saharan Air Layer, the organic matter comes from organic dust (8% of the bulk OM; 63% of the OM

determined) and combustion (6% of the bulk OM; 34% of the OM determined) sources, whereas under the westerlies it comes from organic dust (22% of the bulk OM; 28% of the OM determined), biomass burning (23% of the bulk OM; 49% of the OM determined) and combustion (19% of the bulk OM; 23% of the OM determined) sources, showing that the free troposphere is highly influenced by combustion and biomass burning compounds.

5 Comprehensive knowledge of the organic aerosol chemistry is of great importance in assessing anthropogenic influences and evaluating the effect of radiative forcing. The work presented here offers new insights into the organic composition of the North Atlantic free troposphere as well as the trans-boundary origin of some compounds. Further studies are needed to understand the main mechanisms by which the aerosol is lofted into the free troposphere and transported over long distances.

*Acknowledgements:* This study was performed within the context of the projects AEROATLAN (CGL2015-66299-P; 10 MINECO/FEDER) and TEAPARTICLE (CGL2011-29621), supported by the Ministry of Economy and Competitiveness of Spain and the European Regional Development Fund (ERDF). The authors acknowledge the NOAA Air Resources Laboratory (ARL) for the provision of the HYSPLIT back-trajectories used in this publication. The excellent work performed by the staff of the Atmospheric Research Centre (C. Bayo, C. Hernández, F. de Ory, V. Carreño, R. del Campo and SIELTEC Canarias) and of the Institute of Environmental Assessment and Water Research (R. Chaler, D. Fanjul, and B. 15 Hortelano) is appreciated. M. I. García acknowledges the grant of the Canarian Agency for Research, Innovation and Information Society (ACIISI) co-funded by the European Social Funds. Measurements at Izaña observatory are performed within the context Global Atmospheric Watch networks with the financial support of the State Meteorological Agency of Spain (AEMET).

20 *Competing interests*: The authors declare that they have no conflict of interest.

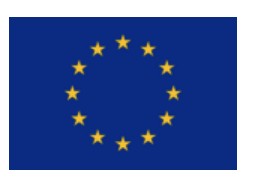
European Regional Development Fund

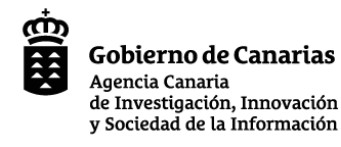

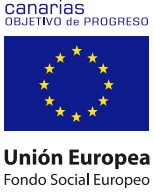

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

**Table 1.** Average concentration of the chemical major compounds for (i) FT-PM$_T$ and BL-PM$_{2.5}$ taking into account all samples, (ii) FT-PM$_T$ and BL-PM$_{2.5}$ collected within the Saharan Air Layer (SAL) , (iii) FT-PM$_T$ and BL-PM$_{2.5}$ collected within the Westerlies (WES) without the FT-PM$_T$ biomass burning event and (iv) FT-PM$_T$ biomass burning event (BBE).

| | FT-PM$_T$ ALL | BL-PM$_{2.5}$ ALL | FT-PM$_T$ SAL | FT-PM$_T$ WES | BL-PM$_{2.5}$ SAL | BL-PM$_{2.5}$ WES | FT-PM$_T$ BBE |
|---|---|---|---|---|---|---|---|
| ∑ CC, µg·m$^{-3}$ | 78.98 | 13.70 | 92.74 | 2.16 | 17.07 | 4.70 | 6.84 |
| Dust, µg·m$^{-3}$ | 73.61 | 11.18 | 86.71 | 1.51 | 14.10 | 3.39 | 1.66 |
| Sea Salt, µg·m$^{-3}$ | 0.53 | 0.36 | 0.59 | 0.27 | 0.25 | 0.66 | < 0.01 |
| OM, µg·m$^{-3}$ | 1.32 | 0.47 | 1.39 | 0.04 | 0.61 | 0.09 | 3.64 |
| EC, µg·m$^{-3}$ | 0.04 | 0.07 | 0.03 | < 0.01 | 0.08 | 0.05 | 0.29 |
| NO$_3^-$, µg·m$^{-3}$ | 0.73 | 0.08 | 0.87 | < 0.01 | 0.11 | < 0.01 | < 0.01 |
| NH$_4^+$, µg·m$^{-3}$ | 0.33 | 0.25 | 0.35 | 0.14 | 0.32 | 0.07 | 0.54 |
| nss-SO$_4^=$, µg·m$^{-3}$ | 2.42 | 1.28 | 2.80 | 0.19 | 1.61 | 0.43 | 0.71 |
| ss-SO$_4^=$, µg·m$^{-3}$ | 0.02 | 0.01 | 0.02 | 0.01 | 0.01 | 0.03 | < 0.01 |

∑**CC**: sum chemical composition; **OM**: organic matter; **EC**: elemental carbon; **nss-SO$_4^=$**: non sea salt sulphate; **ss-SO$_4^=$**: sea salt sulphate.

**Table 2.** Average concentration of the selected organic species for (i) FT-PM$_T$ and BL-PM$_{2.5}$ taking into account all samples, (ii) FT-PM$_T$ and BL-PM$_{2.5}$ collected within the Saharan Air Layer (SAL), (iii) FT-PM$_T$ and BL-PM$_{2.5}$ collected within the Westerlies (WES) without the FT-PM$_T$ biomass burning event and (iv) FT-PM$_T$ biomass burning event (BBE).

| | FT-PM$_T$ ALL | BL-PM$_{2.5}$ ALL | FT-PM$_T$ SAL | FT-PM$_T$ WES | BL-PM$_{2.5}$ SAL | BL-PM$_{2.5}$ WES | FT-PM$_T$ BBE |
|---|---|---|---|---|---|---|---|
| **Levoglucosan** | | | | | | | |
| Levoglucosan, ng·m$^{-3}$ | 0.75 | 0.53 | 0.41 | 0.40 | 0.34 | 1.04 | 9.33 |
| **Dicarboxylic acids** | | | | | | | |
| Succinic, ng·m$^{-3}$ | 6.51 | 3.70 | 5.70 | 3.52 | 4.03 | 2.80 | 33.35 |
| Glutaric, ng·m$^{-3}$ | 1.97 | 0.83 | 1.90 | 0.74 | 0.85 | 0.77 | 7.23 |
| Adipic, ng·m$^{-3}$ | 1.43 | 0.69 | 1.53 | 0.62 | 0.72 | 0.60 | 1.71 |
| Pimelic, ng·m$^{-3}$ | 0.83 | 0.35 | 0.92 | 0.37 | 0.33 | 0.39 | 0.17 |
| Suberic, ng·m$^{-3}$ | 0.48 | 0.30 | 0.50 | 0.29 | 0.29 | 0.31 | 0.39 |
| Azelaic, ng·m$^{-3}$ | 0.88 | 0.79 | 0.93 | 0.57 | 0.78 | 0.82 | 0.71 |
| Malic, ng·m$^{-3}$ | 2.01 | 2.15 | 0.75 | 1.20 | 1.67 | 3.43 | 32.21 |
| Phthalic, ng·m$^{-3}$ | 3.18 | 2.77 | 2.19 | 9.43 | 2.38 | 3.80 | 6.21 |
| **Saccharides** | | | | | | | |
| α-glucose, ng·m$^{-3}$ | 9.26 | 0.90 | 10.79 | 0.62 | 0.90 | 0.89 | 1.65 |
| β-glucose, ng·m$^{-3}$ | 9.13 | 1.00 | 10.63 | 0.61 | 1.01 | 0.98 | 1.65 |
| Fructose, ng·m$^{-3}$ | 2.02 | 1.01 | 2.23 | 0.95 | 1.10 | 0.76 | 0.69 |
| Sucrose, ng·m$^{-3}$ | 2.72 | 0.47 | 3.18 | 0.27 | 0.18 | 1.26 | 0.01 |
| Mannitol, ng·m$^{-3}$ | 0.35 | 0.12 | 0.40 | 0.08 | 0.12 | 0.12 | 0.07 |
| **n-Alkanes** | | | | | | | |
| nC24, ng·m$^{-3}$ | 0.72 | 1.63 | 0.75 | 0.48 | 1.87 | 0.97 | 1.01 |
| nC25, ng·m$^{-3}$ | 0.93 | 2.93 | 0.95 | 0.40 | 3.26 | 2.05 | 2.09 |
| nC26, ng·m$^{-3}$ | 0.60 | 0.64 | 0.65 | 0.26 | 0.69 | 0.49 | 0.54 |
| nC27, ng·m$^{-3}$ | 0.89 | 0.95 | 0.98 | 0.24 | 0.94 | 0.98 | 0.76 |
| nC28, ng·m$^{-3}$ | 0.37 | 0.32 | 0.39 | 0.17 | 0.38 | 0.17 | 0.36 |
| nC29, ng·m$^{-3}$ | 1.18 | 0.42 | 1.34 | 0.25 | 0.48 | 0.25 | 0.63 |
| nC30, ng·m$^{-3}$ | 0.45 | 0.14 | 0.51 | 0.07 | 0.16 | 0.10 | 0.18 |
| nC31, ng·m$^{-3}$ | 1.55 | 0.31 | 1.77 | 0.37 | 0.33 | 0.25 | 0.29 |
| nC32, ng·m$^{-3}$ | 0.39 | 0.10 | 0.45 | 0.05 | 0.11 | 0.06 | 0.08 |
| nC33, ng·m$^{-3}$ | 0.48 | 0.10 | 0.55 | 0.06 | 0.12 | 0.06 | 0.13 |
| nC34, ng·m$^{-3}$ | 0.29 | 0.05 | 0.34 | 0.02 | 0.05 | 0.04 | 0.01 |
| **Hopanes** | | | | | | | |
| Hopane, ng·m$^{-3}$ | 0.06 | 0.02 | 0.07 | 0.01 | 0.03 | 0.01 | 0.03 |
| nor-Hopane, ng·m$^{-3}$ | 0.07 | 0.05 | 0.08 | 0.02 | 0.07 | 0.02 | 0.03 |
| **PAHs** | | | | | | | |
| B [a] A, pg·m$^{-3}$ | 1.48 | 1.48 | 1.58 | 0.80 | 1.61 | 1.13 | 1.37 |
| Chr, pg·m$^{-3}$ | 4.27 | 4.37 | 4.63 | 1.92 | 5.12 | 2.39 | 3.38 |
| B [b+j+k] F, pg·m$^{-3}$ | 3.67 | 5.74 | 4.20 | 0.66 | 6.69 | 3.21 | 1.13 |
| B [e] P, pg·m$^{-3}$ | 1.36 | 1.94 | 1.50 | 0.47 | 2.22 | 1.20 | 0.83 |
| B [a] P, pg·m$^{-3}$ | 0.78 | 1.21 | 0.89 | 0.18 | 1.25 | 1.08 | 0.29 |
| In[123cd] P, pg·m$^{-3}$ | 1.47 | 2.33 | 1.65 | 0.46 | 2.18 | 2.74 | 0.56 |
| B [ghi] Per, pg·m$^{-3}$ | 3.29 | 6.94 | 3.56 | 1.73 | 6.37 | 8.46 | 1.84 |
| **SOA PIN** | | | | | | | |
| Cis-Pinonic, ng·m$^{-3}$ | 27.83 | 15.24 | 32.72 | 0.89 | 13.23 | 20.59 | 1.00 |
| 3-HGA, ng·m$^{-3}$ | 0.21 | 0.51 | 0.09 | 0.24 | 0.39 | 0.81 | 2.88 |
| MBTCA, ng·m$^{-3}$ | 0.03 | 0.24 | 0.01 | 0.05 | 0.13 | 0.54 | 0.27 |
| **SOA ISO** | | | | | | | |
| 2MGA, ng·m$^{-3}$ | 4.22 | 2.38 | 4.46 | 1.63 | 2.65 | 1.65 | 6.56 |
| 2MT-1, ng·m$^{-3}$ | 6.64 | 4.94 | 7.27 | 3.40 | 5.16 | 4.33 | 2.40 |
| 2MT-2, ng·m$^{-3}$ | 15.45 | 9.53 | 16.79 | 8.95 | 9.24 | 10.31 | 5.53 |

**B[a]A**: benz[a]anthracene; **Chr**: chrysene; **B[b+j+k]F**: benzo[b+k]fluoranthene; **B[e]P**: benzo[e]pyrene; **B[a]P**: benzo[a]pyrene; **In[123cd]P**: indeno[1,2,3-cd]pyrene; **B[ghi]Per**: benzo[ghi]perylene; **3-HGA**: 3-hydroxyglutaric acid; **MBTCA**: 3-methyl-1,2,3-butanetricarboxylic acid; **2MGA**: 2-methylglyceric acid; **2MT-1**: 2-methylthreitol; **2MT-2**: 2-methylerythritol.

**Table 3.** Pearson correlation coefficients matrix of the organic and inorganic compounds within the free troposphere ($PM_T$). Statistically significant correlations (p-value < 0.01) are highlighted. BBE was excluded in this analysis.

| | Levoglucosan | Dicarboxylic acids | Saccharides | n-Alkanes | Hopanes | PAHs | SOA PIN | SOA ISO | Dust | Sea Salt | OM | EC | $NO_3^-$ | $NH_4^+$ | nss-$SO_4^=$ | ss-$SO_4^=$ |
|---|---|---|---|---|---|---|---|---|---|---|---|---|---|---|---|---|
| Levoglucosan | 1.0 | | | | | | | | | | | | | | | |
| Dicarboxylic acids | 0.0 | 1.0 | | | | | | | | | | | | | | |
| Saccharides | 0.0 | 0.4 | 1.0 | | | | | | | | | | | | | |
| n-Alkanes | 0.1 | 0.5 | **0.5** | 1.0 | | | | | | | | | | | | |
| Hopanes | 0.0 | **0.6** | 0.7 | **0.8** | 1.0 | | | | | | | | | | | |
| PAHs | 0.3 | 0.3 | 0.2 | 0.3 | 0.3 | 1.0 | | | | | | | | | | |
| SOA PIN | -0.1 | 0.2 | 0.3 | **0.6** | 0.5 | 0.1 | 1.0 | | | | | | | | | |
| SOA ISO | 0.2 | 0.3 | 0.2 | 0.3 | 0.3 | **0.6** | 0.4 | 1.0 | | | | | | | | |
| Dust | 0.0 | **0.7** | 0.6 | **0.7** | **0.9** | 0.2 | **0.7** | 0.2 | 1.0 | | | | | | | |
| Sea Salt | **0.6** | -0.1 | -0.2 | -0.1 | -0.3 | 0.4 | 0.1 | 0.2 | -0.2 | 1.0 | | | | | | |
| OM | -0.1 | 0.3 | 0.3 | **0.7** | **0.8** | 0.2 | **0.8** | 0.4 | **0.9** | 0.0 | 1.0 | | | | | |
| EC | 0.2 | 0.1 | -0.1 | -0.1 | -0.2 | **0.7** | -0.2 | 0.3 | -0.3 | 0.3 | -0.2 | 1.0 | | | | |
| $NO_3^-$ | 0.0 | 0.4 | **0.7** | **0.8** | **0.8** | 0.5 | **0.6** | 0.4 | **0.8** | -0.1 | **0.8** | 0.0 | 1.0 | | | |
| $NH_4^+$ | 0.1 | 0.1 | 0.2 | 0.2 | 0.3 | **0.6** | -0.1 | 0.3 | 0.2 | 0.0 | 0.2 | 0.5 | 0.5 | 1.0 | | |
| nss-$SO_4^=$ | 0.1 | 0.2 | **0.6** | 0.5 | **0.6** | 0.5 | 0.5 | **0.6** | 0.5 | 0.1 | **0.6** | 0.1 | **0.8** | **0.7** | 1.0 | |
| ss-$SO_4^=$ | **0.6** | 0.0 | -0.2 | 0.0 | -0.2 | 0.4 | 0.1 | 0.2 | -0.2 | **1.0** | 0.0 | 0.3 | 0.0 | 0.0 | 0.0 | 1.0 |

**OM**: organic matter; **EC**: elemental carbon; **nss-$SO_4^=$**: non sea salt sulphate; **ss-$SO_4^=$**: sea salt sulphate.

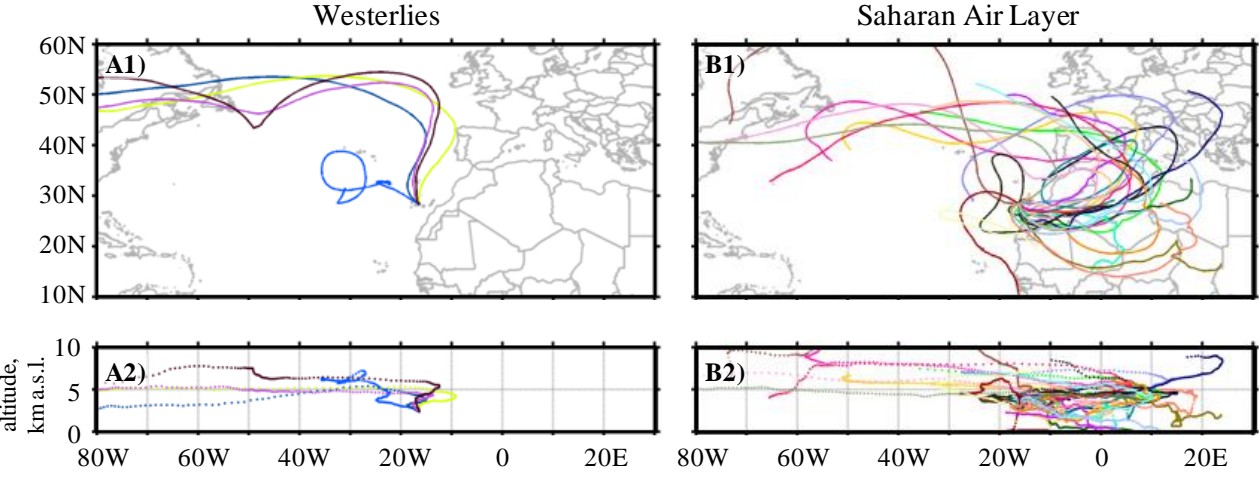

**Figure 1:** Ten-day back-trajectories based on HYSPLIT model for the samples collected within the **(A)** Westerlies (26 Aug–30 Aug) and (**B**) the Saharan Air Layer (01 Aug–25 Aug and 31 Aug–01 Sep); the dates refer to the day of completion of the sampling.

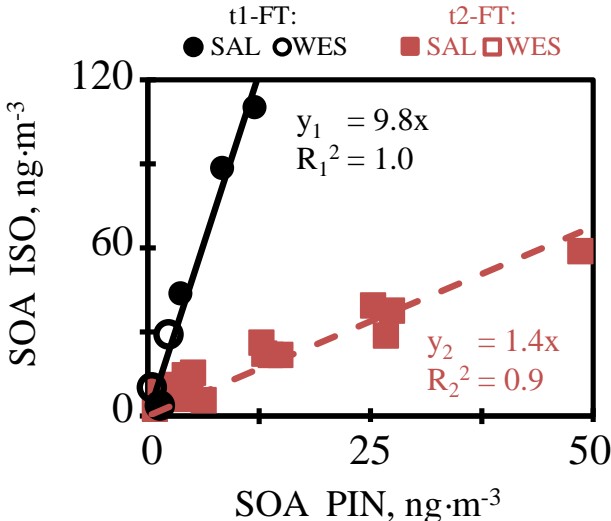

**Figure 2:** Scatter plot between total concentration of SOA ISO and total concentration of SOA PIN within the FT ($PM_T$ collected during the night). Two tendencies can be distinguished: tendency-1 (t1; circles) and tendency-2 (t2; squares). Filled markers correspond to measurements within the SAL and open markers to measurements within the WES.

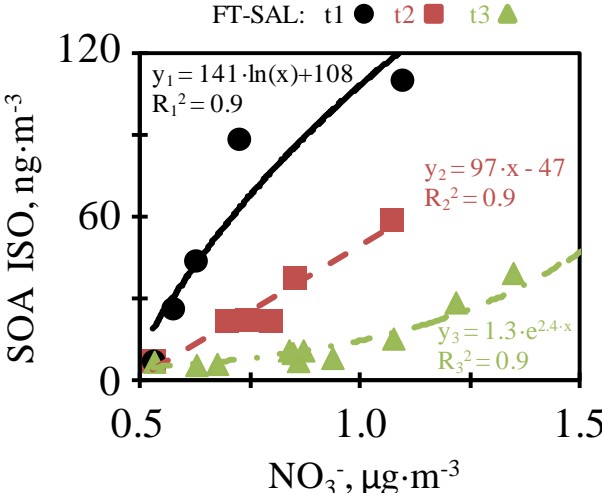

**Figure 3:** Scatter plot between SOA ISO and nitrate within the SAL under FT conditions (FT-PM$_T$ samples collected during the night). Three tendencies can be distinguished: tendency-1 (t1; circles), tendency-2 (t2; squares) and tendency-3 (t3; triangles). Westerlies were excluded when calculating the regression coefficients as values were under the detection limit.

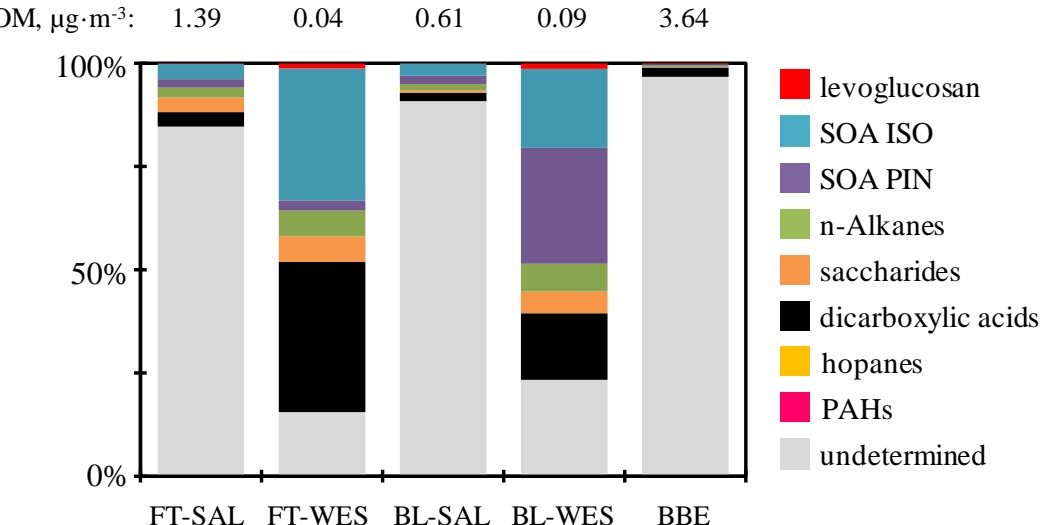

**Figure 4:** Contribution of the eight analysed organic groups to the Izaña OM composition within the FT and the BL under the SAL (FT-SAL and BL-SAL) and the WES (FT-WES, BL-WES, BBE). Average total OM for each air mass is at top. FT-$PM_T$ samples were collected during the night (22–6 GMT) and BL-$PM_{2.5}$ samples were collected during the day (10–16 GMT).

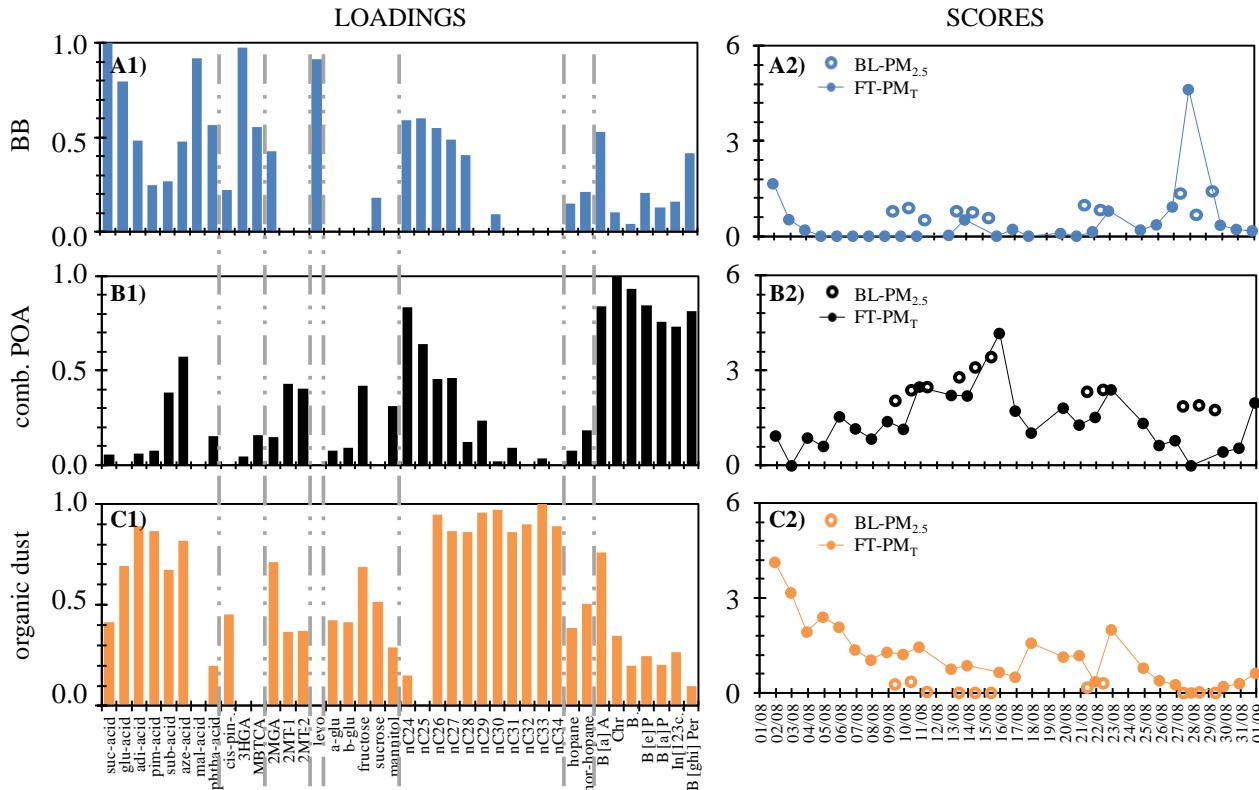

**Figure 5:** FT-PM$_T$ (night) and BL-PM$_{2.5}$ (day) loadings and scores of the three components from MCR-ALS resolved profiles. Filled markers correspond to FT-PM$_T$ and open markers to BL-PM$_{2.5}$. Grey lines separate the compounds belonging to the different organic groups: dicarboxylic acids, SOA PIN, SOA ISO, levoglucosan, saccharides, n-alkanes, hopanes and PAHs (from left to right).

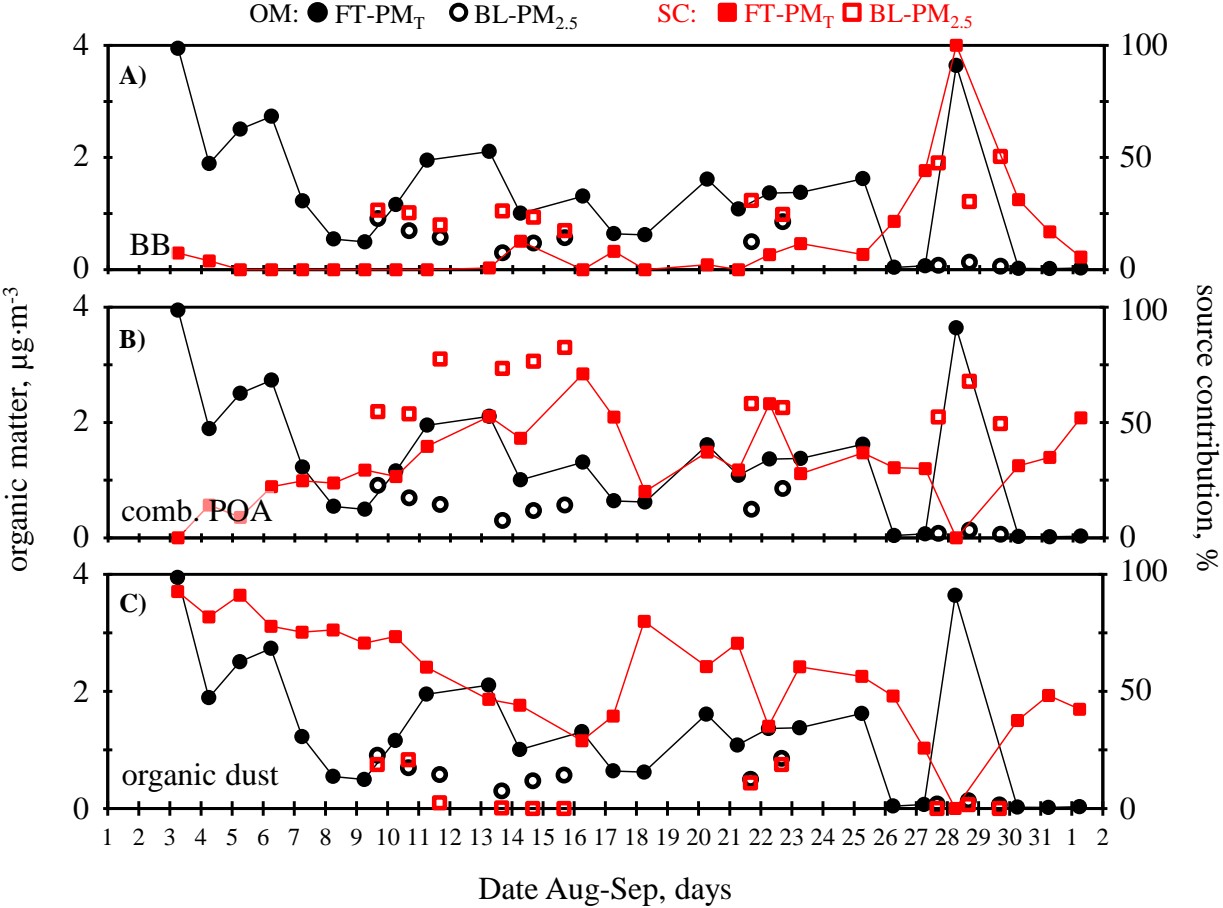

**Figure 6:** Time series of the total organic matter (OM; circle markers) and the source contribution (SC; square markers) to the organic matter determined for the FT-PM$_T$ (filled markets) and the BL-PM$_{2.5}$ samples (open markers). Sources: (**A**) biomass burning (BB), (**B**) combustion (comb.) POA and (**C**) organic dust.

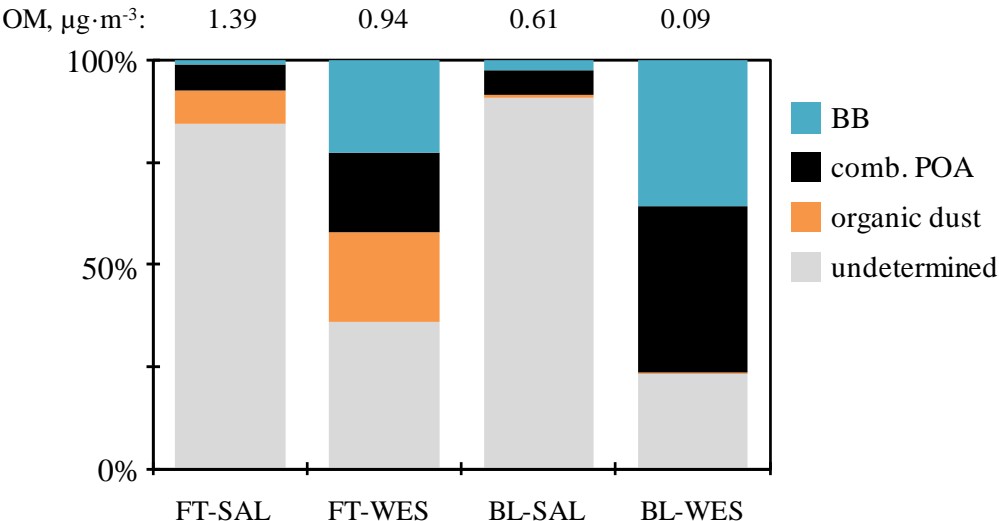

**Figure 7:** Contribution of the identified organic aerosol sources to the total organic matter within the FT and the BL under the SAL (FT-SAL and BL-SAL) and the WES (FT-WES, BL-WES, BBE). Average total OM for each air mass is at top. FT-PM$_T$ samples were collected during the night (22–6 GMT) and BL-PM$_{2.5}$ samples were collected during the day (10–16 GMT).