# Peer review of "Speciation of organic aerosols in the Saharan Air Layer and in the free troposphere westerlies"

_Atmospheric Chemistry and Physics, 2017_

## Referee Comment (RC1) · Anonymous Referee #1 · 27 Mar 2017

This study focused on organic aerosol composition in the free troposphere from airflows transported by weasterlies over the north Atlantic as well as airflows from the Saharan Air Layer. Organic tracers were used to identify possible atmospheric particulate matter sources. Saharan Air was dominated by dust and only contained about 1.5% organic mass. The Westerlies had significantly higher amounts of organic mass fraction, most of which was dicarboxylic acids and isoprene SOA. These measurements highlight the importance of atmospheric transport as an aerosol source.

Minor comment: In figure 2 and 3 you claim that there are multiple tendencies that can reasonably be explained to be from different source locations, however the argument is weak if you do not have some further evidence to support that there are multiple tendencies rather than no correlation when fitting all the data. This is especially important since there are such a few amount of samples. Did you actually find evidence for this

in back trajectories or any other analysis.

---

## Referee Comment (RC2) · Anonymous Referee #2 · 28 Mar 2017

General comment: This manuscript presents a substantial and comprehensive speciation of organic aerosols transported in the SAL and in the North Atlantic free troposphere westerlies. The sampling methods and analysis methods are valid. The Multivariance Curve Resolution Alternating Least Squares (MCR-ALS) model was applied to present sources of organic aerosol. The scientific results are presented in a well-organized way. Atmospheric particulate matter on secondary inorganic species, organic species, elemental composition, chemical composition and 40 organic tracer species were analyzed, which provides possible sources of organic aerosol. Source apportion method also gives the contributions of total organic aerosol from different sources, such as biomass burning (BB), combustion POA, and organic dust. Overall, this manuscript is publishable in Atmospheric Chemistry and Physics with the following minor comments are addressed.

[Figure]

Minor comment:

1.What is " ddmmm" in Figure 1?

2.A large fraction of OM was not determined under FT-SAL, BL-SAL and BBE in Figure 4 or Figure 7. Can authors explain the difference of undetermined fraction between FT-SAL and FT-WES? Is this due to method limitation or the size cut-off of collected particles for dust-associated compounds?

3.P.4 Line 25: Please define "SIM" mode.

4.Can authors explain the "score factor" in the method or in Figure 5?

5.The author wrote the correlations of total concentration of SOA ISO and total concentration of SOA PIN exhibits two distinct trends in Figure 2, and the correlation between SOA ISO and NO3- presents three tendencies. The explanation for these biogenic SOA sources is not clearly discussed and supported by significant evidence. The ratio of isoprene to NOx, and the daytime photooxidation process and nighttime nitrate chemistry can be discussed.

Technical correction:

1.P.16 Line 18: "wad" should be "was".

2.P.6 line 21: Some species "shows" should be "show".

3.P.13 line 9: "represents" should be "represent".

4.P.2 line 14": " An important factor" should be " Some important factors".

---

## Author Comment (AC1) · 24 Apr 2017

**Answer to comment of **Referee#1**

on "*Speciation of organic aerosols in the Saharan Air Layer and in the free troposphere westerlies*" by M.I. García et al.

**Reviewer Comment - OVERVIEW:**

*This study focused on organic aerosol composition in the free troposphere from airflows transported by westerlies over the nor th Atlantic as well as airflows from the Saharan Air Layer. Organic tracers were used to identify possible atmospheric particulate matter sources. Saharan Air was dominated by dust and only contained about 1.5% organic mass. The Westerlies had significantly higher amounts of organic mass fraction, most of which was dicarboxylic acids and isoprene SOA. These measurements highlight the importance of atmospheric transport as an aerosol source.*

**REPLY:**

Thanks to Referee #1 for the useful comments that contribute to improve the original manuscript. Please, find below a point-by-point reply to each question and suggestion.

**Minor comments**

1. In figure 2 and 3 you claim that there are multiple tendencies that can reasonably be explained to be from different source locations, however the argument is weak if you do not have some further evidence to support that there are multiple tendencies rather than no correlation when fitting all the data. This is especially important since there are such a few amount of samples. Did you actually find evidence for this in back trajectories or any other analysis?

   **REPLY:**
   Thank you very much for highlighting this point. The back-trajectories analysis allows identifying air masses from North Africa and North America, but the methodology does not allow a deeper differentiation of potential source regions within North Africa. We do not see differences in the back-trajectories of dusty days associated with high and with low concentrations of nitrate, SOA ISO or SOA PIN, they mostly show the regular circulation associated with the Saharan Air Layer (discussed by Rodríguez et al., 2015, included in the reference list of the ACPD manuscript). The lack of association of a trajectory type with a certain amount of nitrate (or SOA ISO or SOA PIN) is due to the fact that (i) an air mass may have a difference amount of a secondary aerosol depending on the emissions in the source region days backward and (ii) these emissions in the source region change along time.

   In section <3.2.8 Tracers of isoprene oxidation (SOA ISO)> we suggest the correlation between SOA-ISO and SOA-PIN "might be" due to "different global sources of the precursor volatile compounds" as "global estimations of isoprene and α-pinene emissions and sources show they are diverse and not equally distributed in the globe (Luo et al., 2010; Guenther et al., 2012; Sindelarova et al., 2014)". Further studies are needed on this topic.

   We agree the correlation in Fig.3 need more theoretical support. Changes have been introduced in the manuscript.

   **CHANGES IN THE MANUSCRIPT** [R2#C1]:
   We have added the following description to the introduction (italic):
   "Some [R2#C9] important factors influencing SOA formation are reactive nitrogen species *(NO$_x$)* (Presto et al., 2005; Ng et al., 2007, 2008), which are further oxidized to the highly reactive nitrate radical *(NO$_3$). NO$_3$ interacts* with VOCs in gas-phase, likely having an impact in global OA levels as indicated by modelling (Pye et al., 2010) and experimental work (Surratt et al., 2006). *At daytime, NO$_x$ can react with organic peroxy radicals (RO$_2$) resulting in peroxy nitrates (RO$_2$NO$_2$) and alkyl and multifunctional nitrates (RONO$_2$) (O'Brien et al., 1995); the formation of organic nitrates provisionally sequesters NO$_x$, which can suffers long-range transport to more remote environments (Horowitz et al., 1998; Mao et al., 2013). At nighttime, the interaction VOCs-NO$_3$ dominates, with SOA yields greater than that for OH or O$_3$ oxidation (Ng et al., 2016 and references there in). Previous modeling studies carried by Hoyle et al.*

(2007) suggested that, during twilight conditions, ~ 21% of the global average SOA may be due to oxidation of SOA precursors by $NO_3$, and measurements performed by Brown et al. (2009) found that, during nighttime, 1-17% of SOA was the result of $NO_3$ initiated isoprene oxidation."

We have extended the discussion in section <3.2.8 Tracers of isoprene oxidation (SOA ISO)> (italic):
"SOA ISO seems to depend strongly on the conditions (aerosol acidity, $NO_x$ concentrations and *pre-existing aerosol*) used to oxidize isoprene (Surratt et al., 2006, 2007*; Marais et al., 2016). $NO_x$ concentration determine the pathway (low-NOx and high-NOx) followed by the isoprene oxidation, leading to different secondary organic species (Paulot et al., 2009a, b); the low-NOx pathway is ~5 times more efficient than the high-NOx pathway (Marais et al., 2016). Experiments carried out by Kroll et al. (2006) evidence how the isoprene SOA yield varies depending on NOx concentration, increasing from no injected NOx, to a plateau between 100 and 300 ppb NOx, and decreasing at higher NOx concentrations.*

[revised manuscript text omitted]

---

## Author Comment (AC2) · 24 Apr 2017

**Answer to comment of **Referee#2**

on "*Speciation of organic aerosols in the Saharan Air Layer and in the free troposphere westerlies*" by M.I. García et al.

**Reviewer Comment - OVERVIEW:**

*This manuscript presents a substantial and comprehensive speciation of organic aerosols transported in the SAL and in the North Atlantic free troposphere westerlies. The sampling methods and analysis methods are valid. The Multivariance Curve Resolution Alternating Least Squares (MCR-ALS) model was applied to present sources of organic aerosol. The scientific results are presented in a well-organized way. Atmospheric particulate matter on secondary inorganic species, organic species, elemental composition, chemical composition and 40 organic tracer species were analyzed, which provides possible sources of organic aerosol. Source apportion method also gives the contributions of total organic aerosol from different sources, such as biomass burning (BB), combustion POA, and organic dust. Overall, this manuscript is publishable in Atmospheric Chemistry and Physics with the following minor comments are addressed.*

**REPLY:**

Thanks to Referee #2 for the useful comments that contribute to improve the original manuscript. Please, find below a point-by-point reply to each question and suggestion.

**Minor comments**

1. What is " ddmmm" in Figure 1?

   **REPLY:**
   "ddmmm" refer to day and month of sampling. with "ddmmm" referred to ending sampling day.

   **CHANGES IN THE MANUSCRIPT** [R2#C1]**:**
   Sentence
   "with "ddmmm" referred to ending sampling day"
   reworded as
   "*the dates refer to the day of completion of the sampling*".

2. A large fraction of OM was not determined under FT-SAL, BL-SAL and BBE in Figure 4 or Figure 7. Can authors explain the difference of undetermined fraction between FT-SAL and FT-WES? Is this due to method limitation or the size cut-off of collected particles for dust-associated compounds?

   **REPLY:**
   It not due to the size cut-off, since the samples in the FT-SAL and FT-WES airflows were collected in $PM_T$ fraction (i.e. total suspended particles). The organic aerosols comprise thousands of organic species from which only ~10 to 30% has been identified by the scientific community (see details in Andreae, 2009) and the analysis of the samples by means of gas chromatography–mass spectrometry (GC-MS) covers a very small fraction (often < 5 % of the organic matter; Alier et al., 2013, included in the reference list of the ACPD manuscript). The fact that undetermined fraction is higher in the FT-SAL than in the FT-WES indicate that the number of unknown organic species is higher in the SAL than in the WES, so these differences are actually a method limitation. Although the number of identified species is in general low with these techniques, the correlation between the score factors and the organic matter (Table S1) indicates that the selected organic tracers can be representative of the potential sources contributing to the composition of the organic aerosols.

   Reference:
   Andreae, M. O.: A new look at aging aerosols, Science, 326, 1493–1494, doi:10.1126/science.1183158, 2009.

**CHANGES IN THE MANUSCRIPT** [R2#C2]**:**
An explanation of the above mentioned difference of undetermined fraction between FT-SAL and FT-WES has been introduced in "3.2.9 Determined fraction of OM". We have added to the main text: "*Differences in the determined fraction of the samples collected under the FT-SAL and FT-WES influence – as observed in Fig. 4 – is a result of the method limitation as the analysis of the samples by means of gas chromatography–mass spectrometry (GC-MS) covers a very small fraction (often < 5 %; Alier et al., 2013) of the organic matter*".

3. P.4 Line 25: Please define "SIM" mode.

   **REPLY:**
   Thank you very much for your observation.

   **CHANGES IN THE MANUSCRIPT** [R2#C3]**:**
   We have replaced "SIM mode" by "*selected ion monitoring* (SIM) mode".

4. Can authors explain the "score factor" in the method or in Figure 5?

   **REPLY:**
   Thank you very much for your observation.

   **CHANGES IN THE MANUSCRIPT** [R2#C4]**:**
   A description of "loadings" and "scores" has been provided in "3.3 Sources of OM". We have added to the main text: "*The data matrix was decomposed in two factors: loadings (i.e. the relative amount of the chemical compounds in the source) and scores (i.e. the relative contribution of the potential sources to the organic aerosol) (Tauler et al., 2009)*".

   The following reference has been added:
   Tauler, R., Viana, M., Querol, X., Alastuey, A., Flight, R. M., Wentzell, P. D. and Hopke, P. K.: Comparison of the results obtained by four receptor modelling methods in aerosol source apportionment studies, Atmos. Environ., 43(26), 3989–3997, doi:10.1016/j.atmosenv.2009.05.018, 2009.

5. The author wrote the correlations of total concentration of SOA ISO and total concentration of SOA PIN exhibits two distinct trends in Figure 2, and the correlation between SOA ISO and $NO_3^-$ presents three tendencies. The explanation for these biogenic SOA sources is not clearly discussed and supported by significant evidence. The ratio of isoprene to NOx, and the daytime photooxidation process and nighttime nitrate chemistry can be discussed.

   **REPLY:**
   Thank you very much for your suggestion, which will definitively enrich the manuscript. The back-trajectories analysis allows identifying air masses from North Africa and North America, but the methodology does not allow a deeper differentiation of potential source regions within North Africa. We do not see differences in the back-trajectories of dusty days associated with high and with low concentrations of nitrate, SOA ISO or SOA PIN, they mostly show the regular circulation associated with the Saharan Air Layer (discussed by Rodríguez et al., 2015, included in the reference list of the ACPD manuscript). The lack of association of a trajectory type with a certain amount of nitrate (or SOA ISO or SOA PIN) is due to the fact that (i) an air mass may have a difference amount of a secondary aerosol depending on the emissions in the source region days backward and (ii) these emissions in the source region change along time.

   In section <3.2.8 Tracers of isoprene oxidation (SOA ISO)> we suggest the correlation between SOA-ISO and SOA-PIN "might be" due to "different global sources of the precursor volatile compounds" as "global estimations of isoprene and α-pinene emissions and sources show they are diverse and not equally distributed in the globe (Luo et al., 2010; Guenther et al., 2012; Sindelarova et al., 2014)". Further studies are needed on this topic.

We agree that the correlation in Fig.3 needs more theoretical support. Changes have been introduced in the manuscript.

**CHANGES IN THE MANUSCRIPT** [R2#C5]**:**
We have added the following description to the introduction (italic):
"Some [R2#C9] important factors influencing SOA formation are reactive nitrogen species *(NO$_x$)* (Presto et al., 2005; Ng et al., 2007, 2008), which are further oxidized to the highly reactive nitrate radical *(NO$_3$). NO$_3$ interacts* with VOCs in gas-phase, likely having an impact in global OA levels as indicated by modelling (Pye et al., 2010) and experimental work (Surratt et al., 2006). *At daytime, NO$_x$ can react with organic peroxy radicals (RO$_2$) resulting in peroxy nitrates (RO$_2$NO$_2$) and alkyl and multifunctional nitrates (RONO$_2$) (O'Brien et al., 1995); the formation of organic nitrates provisionally sequesters NO$_x$, which can suffers long-range transport to more remote environments (Horowitz et al., 1998; Mao et al., 2013). At nighttime, the interaction VOCs-NO$_3$ dominates, with SOA yields greater than that for OH or O$_3$ oxidation (Ng et al., 2016 and references there in). Previous modeling studies carried by Hoyle et al. (2007) suggested that, during twilight conditions, ~ 21% of the global average SOA may be due to oxidation of SOA precursors by NO$_3$, and measurements performed by Brown et al. (2009) found that, during nighttime, 1-17% of SOA was the result of NO$_3$ initiated isoprene oxidation.*"

We have extended the discussion in section <3.2.8 Tracers of isoprene oxidation (SOA ISO)> (italic):
"SOA ISO seems to depend strongly on the conditions (aerosol acidity, NO$_x$ concentrations and *pre-existing aerosol*) used to oxidize isoprene (Surratt et al., 2006, 2007*; Marais et al., 2016). NO$_x$ concentration determine the pathway (low-NOx and high-NOx) followed by the isoprene oxidation, leading to different secondary organic species (Paulot et al., 2009a, b); the low-NOx pathway is ~5 times more efficient than the high-NOx pathway (Marais et al., 2016). Experiments carried out by Kroll et al. (2006) evidence how the isoprene SOA yield varies depending on NOx concentration, increasing from no injected NOx, to a plateau between 100 and 300 ppb NOx, and decreasing at higher NOx concentrations.*

[revised manuscript text omitted]

**Technical correction**

6. P.16 Line 18: "wad" should be "was".

   **REPLY:**
   Thank you very much for your observation.

   **CHANGES IN THE MANUSCRIPT** [R2#C6]**:**
   We have replaced "wad" by "*was*".

7. P.6 line 21: Some species "shows" should be "show".

   **REPLY:**
   Thank you very much for your observation.

   **CHANGES IN THE MANUSCRIPT** [R2#C7]**:**
   We have replaced "shows" by "*show*".

8. P.13 line 9: "represents" should be "represent".

   **REPLY:**
   Thank you very much for your observation.

   **CHANGES IN THE MANUSCRIPT** [R2#C8]**:**
   We have replaced "represents" by "*represent*".

9. P.2 line 14": "An important factor" should be "Some important factors".

   **REPLY:**

Thank you very much for your observation.

**CHANGES IN THE MANUSCRIPT** [R2#C9]**:**
We have replaced "An important factor" by "*Some important factors*".

---

## Author Response (AR1)

Dear Dr. Russell,

I contact you because of the end of the discussion period of the ms <acp-2017-108>. As you can see in the web site, we have already replied to each of the question of the two reviewers. In the reply we have followed the structure (1) comments from Referees, (2) author's response, (3) author's changes in manuscript, recommended by ACP.

We have now prepared a revised version of the manuscript. For preparing this version we have taken into account the comments of both reviewers. As you can see in the reports, both referees have arisen minor and technical comments and agree publication of this manuscript in ACP. The questions and suggestions of the reviewers have definitively contributed to improve the manuscript.

Please, find attached to this letter (i) the "Authors Response Report", which includes the answers to reviewers 1 and 2, with a reply to each question and the description of the changes in the manuscript as a consequence of each referee comment, and (ii) the "Revised Manuscript and Supplement", where the changes performed (with respect to the ACPD version) are highlighted. We have also followed the structure (1) comments from Referees, (2) author's response, (3) author's changes in manuscript, recommended by ACP, so changes can easily be tracked.

This revised version includes the minor comments and technical changes were introduced to include the comments of the referees.

The latex file of the manuscript and the pdf of the supplement will be uploaded in a further step.

Thanks,
Sergio Rodríguez

**Authors Response Report**

**REPLY TO REVIEWER#1:**

Thanks to Referee #1 for the useful comments that contribute to improve the original manuscript. Please, find below a point-by-point reply to each question and suggestion.

**Minor comments**

1. In figure 2 and 3 you claim that there are multiple tendencies that can reasonably be explained to be from different source locations, however the argument is weak if you do not have some further evidence to support that there are multiple tendencies rather than no correlation when fitting all the data. This is especially important since there are such a few amount of samples. Did you actually find evidence for this in back trajectories or any other analysis?

   **REPLY:**
   Thank you very much for highlighting this point. The back-trajectories analysis allows identifying air masses from North Africa and North America, but the methodology does not allow a deeper differentiation of potential source regions within North Africa. We do not see differences in the back-trajectories of dusty days associated with high and with low concentrations of nitrate, SOA ISO or SOA PIN, they mostly show the regular circulation associated with the Saharan Air Layer (discussed by Rodríguez et al., 2015, included in the reference list of the ACPD manuscript). The lack of association of a trajectory type with a certain amount of nitrate (or SOA ISO or SOA PIN) is due to the fact that (i) an air mass may have a difference amount of a secondary aerosol depending on the emissions in the source region days backward and (ii) these emissions in the source region change along time.

   In section <3.2.8 Tracers of isoprene oxidation (SOA ISO)> we suggest the correlation between SOA-ISO and SOA-PIN "might be" due to "different global sources of the precursor volatile compounds" as "global estimations of isoprene and α-pinene emissions and sources show they are diverse and not equally distributed in the globe (Luo et al., 2010; Guenther et al., 2012; Sindelarova et al., 2014)". Further studies are needed on this topic.

   We agree the correlation in Fig.3 need more theoretical support. Changes have been introduced in the manuscript.

   **CHANGES IN THE MANUSCRIPT** [R2#C1]:
   We have added the following description to the introduction (italic):
   "Some [R2#C9] important factors influencing SOA formation are reactive nitrogen species *(NO$_x$)* (Presto et al., 2005; Ng et al., 2007, 2008), which are further oxidized to the highly reactive nitrate radical *(NO$_3$). NO$_3$ interacts* with VOCs in gas-phase, likely having an impact in global OA levels as indicated by modelling (Pye et al., 2010) and experimental work (Surratt et al., 2006). *At daytime, NO$_x$ can react with organic peroxy radicals (RO$_2$) resulting in peroxy nitrates (RO$_2$NO$_2$) and alkyl and multifunctional nitrates (RONO$_2$) (O'Brien et al., 1995); the formation of organic nitrates provisionally sequesters NO$_x$ which can suffers long-range transport to more remote environments (Horowitz et al., 1998; Mao et al., 2013). At nighttime, the interaction VOCs-NO$_3$ dominates, with SOA yields greater than that for OH or O$_3$ oxidation (Ng et al., 2016 and references there in). Previous modeling studies carried by Hoyle et al. (2007) suggested that, during twilight conditions, ~ 21% of the global average SOA may be due to oxidation of SOA precursors by NO$_3$, and measurements performed by Brown et al. (2009) found that, during nighttime, 1-17% of SOA was the result of NO$_3$ initiated isoprene oxidation.*"

   We have extended the discussion in section <3.2.8 Tracers of isoprene oxidation (SOA ISO)> (italic):
   "SOA ISO seems to depend strongly on the conditions (aerosol acidity, NO$_x$ concentrations and *pre-existing aerosol*) used to oxidize isoprene (Surratt et al., 2006, 2007; *Marais et al., 2016). NO$_x$ concentration determine the pathway (low-NOx and high-NOx) followed by the isoprene oxidation, leading to different secondary organic species (Paulot et al., 2009a, b); the low-NOx pathway is ~5 times more efficient than the high-NOx pathway (Marais et al., 2016). Experiments carried out by Kroll et al. (2006) evidence how the isoprene SOA yield varies depending on NOx concentration, increasing*

[revised manuscript text omitted]

**REPLY TO REVIEWER#2:**

Thanks to Referee #2 for the useful comments that contribute to improve the original manuscript. Please, find below a point-by-point reply to each question and suggestion.

**Minor comments**

**1.** What is " ddmmm" in Figure 1?

**REPLY:**
"ddmmm" refer to day and month of sampling. with "ddmmm" referred to ending sampling day.

**CHANGES IN THE MANUSCRIPT** [R2#C1]**:**
Sentence
 "with "ddmmm" referred to ending sampling day"
reworded as
 "*the dates refer to the day of completion of the sampling*".

**2.** A large fraction of OM was not determined under FT-SAL, BL-SAL and BBE in Figure 4 or Figure 7. Can authors explain the difference of undetermined fraction between FT-SAL and FT-WES? Is this due to method limitation or the size cut-off of collected particles for dust-associated compounds?

**REPLY:**
It not due to the size cut-off, since the samples in the FT-SAL and FT-WES airflows were collected in $PM_T$ fraction (i.e. total suspended particles). The organic aerosols comprise thousands of organic species from which only ~10 to 30% has been identified by the scientific community (see details in Andreae, 2009) and the analysis of the samples by means of gas chromatography–mass spectrometry (GC-MS) covers a very small fraction (often < 5 % of the organic matter; Alier et al., 2013, included in the reference list of the ACPD manuscript). The fact that undetermined fraction is higher in the FT-SAL than in the FT-WES indicate that the number of unknown organic species is higher in the SAL than in the WES, so these differences are actually a method limitation. Although the number of identified species is in general low with these techniques, the correlation between the score factors and the organic matter (Table S1) indicates that the selected organic tracers can be representative of the potential sources contributing to the composition of the organic aerosols.

    **REPLY:**
    Thank you very much for your observation.

    **CHANGES IN THE MANUSCRIPT** [R2#C9]**:**
    We have replaced "An important factor" by "*Some important factors*".

**Revised Manuscript**

Changes are highlighted in green for reviewer#1, in yellow for reviewer#2 and in blue for authors. Brackets indicate the reviewer and comment that prompt the change.

[revised manuscript text omitted]

**Revised Supplement**

Changes are highlighted in green for reviewer#1, in yellow for reviewer#2 and in blue for authors. Brackets indicate the reviewer and comment that prompt the change.

Changes were not needed in the supplement.

---

## Author Response (AR2)

Dear Dr. Russell,

Thank you very much for your observations on the manuscript <acp-2017-108>. We have already replied to each of your questions following the structure (1) comments from Editor, (2) author's response, (3) author's changes in manuscript, recommended by ACP.

We have now prepared a revised version of the manuscript. For preparing this version we have taken into account your comments. In addition, the manuscript has been reviewed by a professional translator, that you can contact if you think is necessary (Elaine Ann Barbour, elaineannbarbour@gmail.com).

Please, find attached (i) the "Authors Response Report", which includes the answer to Editor, with a reply to each question and the description of the changes in the manuscript as a consequence of each comment, (ii) the "Revised Manuscript" and (iii) the "Revised Supplement", where the changes performed (with respect to the version uploaded after the reviewers comments) are highlighted, so changes can easily be tracked.

This revised version includes the Editor minor comments and correction of language/spelling problems.

The latex file of the manuscript and the pdf of the supplement will be uploaded in a further step.

Thanks,
Sergio Rodríguez

**Authors Response Report**

**REPLY TO EDITOR Dr. Russell**

Please, find below a point-by-point reply to each question and suggestion.

**1.** The following partial list of the Language/spelling problems that remain in the revised manuscript and serve to obfuscate the science intended:

    Aerosol cocktail
    The determined organic
    At daytime
    Can suffers
    Mounts
    As is >> and is
    The arriving
    Meet cooking
    Up whirling
    Guenter
    Monoterpenes reacts
    Than this found >> than that found
    A same origin thought the same
    Is pointed as
    Arriving to
    Could also contributes with
    Concentration determine
    Associated to
    Implication on
    "a" 15%, 84%, etc.
    "This determined fraction" means a fraction that has made up its mind not a fraction that has been determined.
    Accounted by
    Significantly representative of
    Predominant presence
    Those form fungi
    Because of
    Highly significant
    Undermined >> undetermined???
    Determined OM

Since the above list of syntax issues is not complete, I ask that you provide a signed certification by a native English speaker (or equivalent) that the revised paper has been corrected. Alternatively you hire ACP to do language editing, since this job should not fall either to reviewers or associate editors.

**REPLY:**
Thank you very much for your deep revision. We really apologize for all this language/spelling problems, which we will avoid in future submitted articles. The changes you have suggested have been specifically pointed as [Ed#C1]. All manuscript has been revised by a professional translator (highlighted in green).

**CHANGES IN THE MANUSCRIPT** [Ed#C1]**:**

| | | |
|---|---|---|
| "Aerosol cocktail" | >> | Aerosols |
| "The determined organic" | >> | The organic fraction determined |
| "At daytime" | >> | In daytime |
| "Can suffers" | >> | Can suffer |
| "Mounts" | >> | Mountains |
| "As is" | >> | and is |
| "The arriving" | >> |  "the air mass started moving towards.." |
| "Meet cooking" | >> | Meat cooking |
| "Up whirling" | >> | Up-lifted |
| "Guenter" | >> | Guenther |
| "Monoterpenes reacts" | >> | Monoterpenes react |
| "Than this found" | >> | than that found |
| "A same origin thought the same | >> | Sentence: "…indicates a same origin thought the same …" Reworded as: "…would seem to indicate they formed through the same…" |
| "Is pointed as" | >> | Sentence: "…Europe is pointed as a potential source …" Reworded as: "…Europe is a potential source …" |
| "Arriving to" | >> | "Arriving at" |
| "Could also contributes with" | >> | "may also contribute with" |
| "Concentration determine" | >> | Concentration determines |
| "Associated to" | >> | Associated with |
| "Implication on" | >> | Implications for |
| "a" 15%, 84%, etc. | >> | "a" deleted from list in <3.2.9 Determined fraction of OM> and other % |
| "This determined fraction" means a fraction that has made up its mind not a fraction that has been determined. | >> | "determined fraction" changed by "the fraction determined" inthe manuscript |
| "Accounted by" | >> | Accounted for |
| "Significantly representative of" | >> | representative of |
| "Predominant presence" | >> | Major presence |
| "Those form fungi" | >> | Those related to fungi |
| "Because of" | >> | because |
| "Highly significant" | >> | statistically significant |
| "Undermined" | >> | undetermined |
| "Determined OM" | >> | "Fraction determined of OM" |

**2.** Also, claims for significance and correlation should be related to a standard benchmark; specify yours and apply it consistently throughout, defining all phrases such as "significantly", "highly representative", etc.

**REPLY:**
Thanks for your suggestion, which definitively will improve the final revised manuscript. We have specified in <3.2 Organic molecular tracers> that (i) the correlations among the organic groups and the major species are evaluated by means of the Pearson correlation coefficient (r) and that (ii) the significance levels are determined according to the p-value. We have replaced the coefficient of correlation ($R^2$) by the Pearson correlation coefficient (r) thought the text and apply this coefficient it consistently throughout phrases such as "statistically significant correlation".

**CHANGES IN THE MANUSCRIPT** [Ed#C2]**:**

1. Sect. <3.2 Organic molecular tracers>
   Sentence:
   "In order to improve the insight on the origin and sources of some FT organic aerosol, correlations among the organic groups and the major species are evaluated (Table 3)."
   Reworded as:
   "In order to improve the insight on the origin and sources of some FT organic aerosol, correlations among the organic groups and the major species are evaluated by means of the Pearson correlation coefficient (r) (Table 3); significance levels are determined according to the p-value (p). This coefficient is applied for all correlation thought the manuscript. "

2. Sect. <3.2.3 Saccharides>
   Sentence:
   "…both isomers showed a good correlation ($R^2$= 0.998)…"
   Reworded as:
   "…both isomers showed a statistically significant correlation (r = 0.99, p<0.01)…"

3. Sect. <3.2.4 n-Alkanes>
   Sentence:
   "…is a significant correlation between…"
   Reworded as:
   "…is a statistically significant correlation between…"

4. Sect. <3.2.5 Hopanes>
   Sentence:
   "The quantified hopane and norhopane showed very good linear correlations (R2= 0.95), pointing to the same emission sources."
   Reworded as:
   "The quantified hopane and norhopane showed a statistically significant correlation (r = 0.97, p<0.01), implying the same emission sources."

   Sentence:
   "A strong correlation is observed in the FT…"
   Reworded as:
   "A statistically significant correlation is observed in the FT…"

5. Sect. <3.2.6 Polycyclic aromatic hydrocarbons>
   Sentence:
   "In the FT there is a strong correlation between PAHs and EC (r = 0.7, p<0.01…"
   Reworded as:
   "In the FT there is a statistically significant correlation between PAHs and EC (r = 0.7, p<0.01…"

6. Sect. <3.2.7 Tracers of α-pinene oxidation (SOA PIN)>
   Sentence:
   "…with a significant correlation ($R^2$ = 0.81) that points…"
   Reworded as:
   "…with a statistically significant correlation (r = 0.90, p<0.01) pointing…"

7. Sect. <3.2.8 Tracers of isoprene oxidation (SOA ISO)>
   Sentence:
   "A significant correlation among 2-MT1 and 2-MT2 was found for individual values (R2 = 0.94)…"
   Reworded as:
   "A statistically significant correlation among 2-MT1 and 2-MT2 was found for individual values (r = 0.90, p<0.01)…"

   Sentence:
   "This significant correlation between …"
   Reworded as:
   "This statistically significant correlation between …"

Sentence:
"…the same temporal trend (FT-SAL 2-MTs us 2-MGA-r2: 0.1), which…"
Reworded as:
"…the same temporal trend (r = 0.4, p<0.05 within the FT-SAL), which…"

Sentence:
"Significant…"
 Reworded as:
"Statistically significant…"

8.  Sect. <3.3.1 Biomass Burning (BB)>
    Sentence:
    "…significantly associated to…"
     Reworded as:
    "… significantly correlated to…"

9.  Sect. <3.3.2 Combustion POA>
    Sentence:
    "…and highly representative for the BL, as shown by its Pearson correlation coefficient…"
    Reworded as:
    "…and representative for the BL, as shown by its statistically significant correlation…"

10. Sect. <3.3.3 Organic dust>
    Sentence:
    "The scores of this component display the highest correlation with dust…"
     Reworded as:
    "The scores of this component display the highest statistically significant correlation with dust…"

11. Sect. <3.3.4 Source apportionment of OM in the SAL and the westerlies>
    Sentence:
    "The correlation between the sum of the three components scores and the OM within the FT (OM r-FT = 0.63; Table S1) indicates that the identified sources might describe, not only the determined fraction of the OM but also the total OM. This significant correlation is not seen for the BL (OM r-BL = 0.33; Table S1)…"
    Reworded as:
    "The statistically significant correlation between the sum of the three components scores and the OM within the FT (OM r-FT = 0.63, p < 0.05; Table S1) indicates that the identified sources might describe, not only the determined fraction of the OM but also the total OM. This significant correlation is not seen for the BL (OM r-BL = 0.33, p < 0.05; Table S1)…"

 3.  Recommend Cite Garrett et al. 2003 JGR

**REPLY:**
Thank you very much for recommending us this valuable study. We have added a reference to the study of Garrett et al. (2003) in section <3.2.8 Tracers of isoprene oxidation (SOA ISO)>

**CHANGES IN THE MANUSCRIPT** [Ed#C3]**:**

Sentence added:
"; this is supported by a previous study (Garrett et al., 2003) in which it was suggested that carbonaceous, sulphate, and nitrate particles – in aerosol plumes transported from North Africa over the North Atlantic Ocean within the FT – were anthropogenic pollution from Europe."

Reference added:
Garrett, T. J., Russell, L. M., Ramaswamy, V., Maria, S. F., and Huebert, B. J.: Microphysical and radiative evolution of aerosol plumes over the tropical North Atlantic Ocean, J. Geophys. Res. Atmos., 108, 4022–4038, doi: 10.1029/2002JD002228, 2003.

**Revised Manuscript**

Changes are highlighted in yellow (based on Editor comments) and green (language corrections). Brackets indicate the comment that prompts the change.

[revised manuscript text omitted]